# Quadratic Upper Bound for Boosting Robustness

**Euijin You** [1]   **Hyang-Won Lee** [1]

## Abstract

Fast adversarial training (FAT) aims to enhance the robustness of models against adversarial attacks with reduced training time, however, FAT often suffers from compromised robustness due to insufficient exploration of adversarial space. In this paper, we develop a loss function to improve robustness in FAT without requiring stronger inner maximization. Specifically, we derive a quadratic upper bound (QUB) on the adversarial training (AT) loss function and propose to utilize the bound with existing FAT methods. Our experimental results show that applying QUB loss to the existing methods yields significant improvement of robustness. Furthermore, using various metrics, we demonstrate that this improvement is likely to result from the smoothened loss landscape of the resulting models.

## 1. Introduction

Deep neural networks have shown remarkable performance in various tasks like image classification and speech recognition, bringing innovation in real-world applications. However, these models have been found to be vulnerable to adversarial attacks (Goodfellow et al., 2015). These attacks involve adding small perturbations to the data to mislead the model into making incorrect predictions, severely compromising its reliability.

Adversarial training (AT) (Madry et al., 2018) is the most common defense method to address this issue. It generates adversarial examples by perturbing the input data during the training process and trains the model with these examples to improve robustness. Accordingly, the robustness of the model obtained by AT depends heavily on the quality of generated adversarial examples. A well-known approach, projected gradient descent (PGD) (Madry et al., 2018), is a

gradient-based iterative method that updates perturbations in the direction that maximizes the loss to explore stronger attacks. There are several other approaches that generate adversarial examples by introducing more sophisticated loss functions (Zhang et al., 2019; Wang et al., 2019). While these approaches can effectively enhance the robustness, they often require time-consuming and computationally expensive training due mainly to the need for generating high-quality examples and performing iterative optimization.

To address this issue, the concept of fast adversarial training (FAT) has emerged (Shafahi et al., 2019). FAT utilizes single-step attacks instead of multi-step attacks, enabling faster and more cost-efficient training, and hence, allows for only a coarse-grained exploration of the adversarial space. As a result, FAT often suffers from catastrophic overfitting (Wong et al., 2020), where the model becomes excessively robust to the adversarial examples encountered during training while compromising robustness against unseen attacks.

Various FAT methods have since been proposed to overcome catastrophic overfitting and improve performance (Andriushchenko & Flammarion, 2020; Kim et al., 2021; Sriramanan et al., 2020; 2021; Jia et al., 2022a). By analyzing the causes of the problem and approaching it from new perspectives, these methods have demonstrated improved robustness, even with reduced training time.

In this paper, we propose a simple loss function designed to enhance robustness without significantly affecting the short training time of FAT. We derive a quadratic upper bound (QUB) on the AT loss function by relying on the fact that the cross-entropy loss function is convex with respect to logits. The bound can indeed be viewed as a combination of the terms for accounting for robust accuracy as well as standard accuracy. We propose to utilize the bound for adversarial training instead of the original AT loss function. Experimental results show that our QUB can further enhance the robustness of existing AT methods without significantly increasing training time. We also analyze various metrics and demonstrate that employing QUB has the effect of smoothing the loss landscape with respect to perturbations, which contributes to enhanced robustness.

The contributions of this paper can be summarized as follows:

---

[1]Department of Computer Science and Engineering, Konkuk University, Seoul, South Korea. Correspondence to: Hyang-Won Lee <leehw@konkuk.ac.kr>.

*Proceedings of the 42^{nd} International Conference on Machine Learning*, Vancouver, Canada. PMLR 267, 2025. Copyright 2025 by the author(s).

- **Proposal of QUB Loss**: We derive a quadratic upper bound (QUB) on the adversarial training loss function by using the convexity of cross-entropy loss with respect to logits.

- **FAT Method with QUB Loss**: We develop an FAT method with QUB loss that can improve model robustness by simply replacing the loss function.

- **Experimental Performance Validation**: We do not only demonstrate improved robustness on various dataset but also analyze the impact of our QUB loss on the loss landscape using various metrics.

## 2. Related Work

### 2.1. Adversarial Attack

Adversarial Attack refers to the method of adding small perturbations to the input data that intentionally degrade the model's performance. In this paper, we focus on attacks in image classification. In the following, $f_\theta(\cdot)$ refers to an artificial neural network with parameter $\theta$, and $\mathcal{L}$ denotes the loss function. The adversarial attack aims to find a perturbation, denoted as $\delta$, that maximizes the so-called AT loss function as follows:

$$\max_{\|\delta\|_p \leq \epsilon} \mathcal{L}(f_\theta(x + \delta), y), \tag{1}$$

where $L_p$ norm constraint ensures that the perturbed input $x+\delta$ lies within the $\epsilon$-ball $B(x, \epsilon)$ centered at the input $x$. In many studies, the $L_\infty$-norm is typically adopted, as it allows for the widest range of attacks (or perturbations) for the same value of $\epsilon$. Since this optimization problem (1) is non-convex, finding the exact optimal solution is very difficult. Therefore, gradient-based methods have been proposed to approximate the solution.

The fast gradient sign method (FGSM) (Goodfellow et al., 2015) is a simple gradient-based attack that adjusts $x$ in the direction of the gradient of the loss as follows:

$$x_{\text{adv}} = x + \epsilon \cdot \text{sign}(\nabla_x \mathcal{L}(f_\theta(x), y)). \tag{2}$$

FGSM generates adversarial examples with a single gradient calculation, allowing attacks to be performed in a short time. However, it has the limitation of adding perturbations by a fixed amount $\epsilon$, thus failing to sufficiently explore all potential attack points within the $\epsilon$-ball.

Projected gradient descent (PGD) (Madry et al., 2018) is an iterative gradient-based method that generates stronger attacks, providing a direct approach to solve problem (1). In the $t$-th iteration, the adversarial example $x_t$, representing the perturbed image after $t$ iterations, is updated as follows:

$$x_{t+1} = \Pi_{\mathcal{B}(x_t, \epsilon)}(x_t + \alpha \cdot \text{sign}(\nabla_x \mathcal{L}(f_\theta(x_t), y))), \tag{3}$$

where $\Pi_{\mathcal{B}(x_t, \epsilon)}(\cdot)$ denotes the projection onto the $\epsilon$-ball around $x_t$. In this method, the image is iteratively updated with a small step size $\alpha$. If the resulting perturbation exceeds the $\epsilon$-bound, it is projected back onto the $\epsilon$-ball to ensure that the constraints are satisfied. PGD obviously generates more powerful attacks than FGSM due to its iterative updates, which potentially makes it to converge to the point where the loss is maximized.

### 2.2. Adversarial Training

To enhance robustness against adversarial attacks, adversarial training was proposed, where adversarial examples are generated and used to train the model (Madry et al., 2018). The min-max optimization problem of adversarial training is written as follows:

$$\min_\theta \max_{\|\delta\|_p \leq \epsilon} \mathcal{L}(f_\theta(x + \delta), y), \tag{4}$$

where the inner maximization seeks perturbations that maximize the loss, and the outer minimization updates the model parameters to minimize this worst-case loss. This approach aims to train the model to make accurate predictions even when it encounters perturbed inputs. By doing so, the model can achieve robustness not only in clean image classification but also in the presence of intentional attacks. Later, TRADES (Zhang et al., 2019), which aims to align the model's output distributions before and after the attack, demonstrated superior robustness performance.

### 2.3. Fast Adversarial Training

In adversarial training, perturbations must be generated for all training data to train the model. However, multi-step update processes, such as PGD, require substantial computational resources and can lead to excessively longer training times. To address this issue, fast adversarial training (FAT) methods have been developed, which generate perturbations in a single step for faster training, such as FGSM in (2).

However, as demonstrated in (Madry et al., 2018), the models trained with FGSM exhibit a lack of robustness to PGD attacks. This is attributed to FGSM's reliance on a single-step update, which proves inadequate for identifying effective attack points. Free-AT (Shafahi et al., 2019) aims to replicate the iterative nature of PGD by generating adversarial examples through a single backward pass, efficiently combining the robustness of iterative attacks with reduced computational overhead. FGSM-RS (Wong et al., 2020) simplifies Free-AT's approach by incorporating random starts, which enables to explore a more diverse perturbation space. N-FGSM (de Jorge Aranda et al., 2022) strengthens single-step adversarial training by using stronger noise and removing gradient clipping, which prevents catastrophic overfitting.

Furthermore, many of the aforementioned FAT methods can suffer from catastrophic overfitting, which is a phenomenon that the model's robustness dramatically decreases against PGD attacks during training. This occurs when the model becomes excessively sensitive to adversarial attacks or overfits to specific types of perturbation (Kurakin et al., 2017; Tramèr et al., 2018; Lin et al., 2024b). As a result, the model may successfully defend against the attack types it was trained on but loses defense capability against more advanced attacks generated in different ways. Several improved FAT methods have been proposed to resolve this issue, and we discuss those methods in the following.

**Regulating Attack Intensity with Step Size Adjustment**

FGSM-CKPT (Kim et al., 2021) avoids direct updates with large step sizes in the adversarial direction, and instead introduces checkpoints to generate better attacks with varying step sizes. ATAS (Huang et al., 2023) identifies fixed step sizes as a cause of catastrophic overfitting and addresses this by introducing an adaptive step size, which adjusts the perturbation size based on the loss variation.

**Enhancing the Initial Points of Attacks**

The robustness of the trained model depends heavily on the initial point of attack, and consequently, several papers have explored improved initialization techniques. FGSM-SDI (Jia et al., 2022b) uses a generator to create perturbations in conjunction with the model, enabling the learning of adversarial attacks tailored to each sample. FGSM-PGI (Jia et al., 2022a) and FGSM-PGK (Jia et al., 2024b) are representative of prior-guided initialization methods, where perturbations from previous epochs are stored and used as initial points for attacks in the next epoch, achieving better performance with only a single step. FGSM-UAP (Pan et al., 2024) proposes a method using a small number of Universal Adversarial Perturbations (UAPs), saving memory while performing strong attacks without storing individual perturbations for whole images.

**Integrating Regularizers into Loss Functions**

There are many studies that seek to enhance the stability and robustness of adversarial training through **regularizers** in the loss function.

FGSM-GA (Andriushchenko & Flammarion, 2020) improves robustness by maximizing the cosine similarity, $\cos(\nabla_x f(x), \nabla_x f(x + \eta))$, where $\eta$ is a random perturbation, promoting consistent gradient alignment between clean and adversarial examples. Guided Adversarial Training (GAT) (Sriramanan et al., 2020) introduces a relaxation term in the basic loss function to find better gradient directions, improving attack efficiency and overall performance. NuAT (Sriramanan et al., 2021) used a nuclear-norm regularizer to optimize by leveraging joint statistics within a mini-batch, rather than processing each data sample independently. A regularizer was designed to align the perturbation direction of FGSM with the loss gradient direction, compensating for the drawbacks of FGSM while achieving performance close to PGD-based training. FAT-CS (Zhao et al., 2023) proposed a regularizer to stabilize the loss convergence process, addressing the issue of catastrophic overfitting, which is often accompanied by sudden changes in loss. FGSM-LAW (Jia et al., 2024a) introduced Lipschitz regularization and auto weight averaging methods to comprehensively improve the model's robustness. Layer-Aware Adversarial Weight Perturbation (LAP) (Lin et al., 2024c) analyzed the varying degree of distortion across layers and applied adaptive weight perturbations to different layers to enhance robustness. ELLE (Rocamora et al., 2024) refines local linearity regularization, with ELLE reducing computational overhead by linking regularization to loss curvature.

**Other Approaches**

Sub-AT (Li et al., 2022) enhances adversarial training by extracting subspaces in the latent space. Dropout scheduling (Vivek & Babu, 2020) prevents gradient masking, while ZeroGrad (Golgooni et al., 2023) removes weak perturbations to improve learning. DOM (Lin et al., 2024a) mitigates over-memorization by adjusting high-confidence predictions. SLAT (Park & Lee, 2021) normalizes feature gradients by applying perturbations in the latent space.

Many of the previous works (using the original AT loss function) have focused on the generation of sophisticated adversarial examples or on the design of regularizers for boosting robustness. Our approach differs from those AT methods in that we derive an upper bound on the AT loss function and use the bound to explore robust models. Furthermore, our method can be readily applied to the existing methods that rely on the AT loss function in (4).

## 3. Quadratic Upper Bound for AT

In this section, we derive a quadratic upper bound on the loss function $\mathcal{L}(f_\theta(x + \delta), y)$ in (4) which is commonly used in adversarial training. In the following, the function $\mathcal{L}(f_\theta(x + \delta), y)$ is denoted as $\mathcal{L}_{\text{AT}}$ and called the AT loss function. Conceivably, if there is an upper bound on the AT loss function which provides a good approximation without losing too much of the characteristics of $\mathcal{L}_{\text{AT}}$, then the adversarial training problem in (4) or its variants might be approximately solved using the upper bound. Our bound indeed captures the key characteristics of $\mathcal{L}_{\text{AT}}$ (will be discussed later), and hence can be used for adversarial training.

## 3.1. Deriving the Bound

The AT loss function is nothing but the cross-entropy loss function with perturbed input, and we begin with the definition of the cross-entropy loss function. Consider the classification problem with $C$ classes. Let $z_i$ denote the logit corresponding to the $i$-th class, where $i \in [1, ..., C]$. The cross-entropy loss function is written as

$$\mathcal{L}(z, y) = -\sum_{i=1}^{C} y_i \log(\hat{y}_i), \quad \hat{y}_i = \frac{e^{z_i}}{\sum_{j=1}^{C} e^{z_j}}, \quad (5)$$

where $y_i$ is the $i$-th element of the one-hot encoded vector $y$, and $\hat{y}$ represents the softmax probabilities of the model's output. This function is known to be convex with respect to the logit vector $z = [z_i, i = 1, ..., C]$ (See Appendix A).

Let us introduce the notation $f$ which overrides the logit vector $z$. We also use $f_\theta$ to indicate that $f_\theta$ represents the logit vector produced by the neural network parameterized by $\theta$. Similarly, $f(x)$ or $f_\theta(x)$ indicates the logit when the input is $x$. The cross-entropy loss function can then be written as $\mathcal{L}(f(x), y)$. However, for convenience, we simplify the notation as $\mathcal{L}(f(x))$, omitting the explicit dependence on $y$. With this notation, the AT loss function is written as $\mathcal{L}(f(x + \delta))$. The following lemma provides an upper bound on the AT loss function.

**Lemma 1.** *The AT loss function is upper-bounded as follows:*

$$\mathcal{L}(f(x+\delta)) \leq \mathcal{L}(f(x)) + (f(x+\delta) - f(x))^T \nabla_f \mathcal{L}(f(x))$$
$$+ \frac{\|\boldsymbol{H}\|_2}{2} \|f(x+\delta) - f(x)\|_2^2, \quad (6)$$

*where $\nabla_f \mathcal{L}$ is the gradient of the loss with respect to the logit $f$ and $\|\boldsymbol{H}\|_2$ is the $L_2$ norm of the Hessian matrix of the loss with respect to the logit, evaluated at some point between $f(x)$ and $f(x + \delta)$.*

The proof of this lemma uses the convexity of cross-entropy loss function $\mathcal{L}(f(x))$ with respect to $f(x)$, and the details can be found in Appendix B. This bound is quadratic with respect to the perturbed logit vector $f(x + \delta)$, and hence, we call the bound quadratic. In order to use the bound in adversarial training, we need to specify the value $\|\boldsymbol{H}\|_2$ or its upper bound, which is given in the following lemma.

**Lemma 2.** *We have $\|\boldsymbol{H}\|_2 \leq \frac{1}{2}$.*

The derivation of the bound is presented in Appendix C.

Based on Lemmas 1 and 2, the QUB loss is defined as

$$\mathcal{L}_{\text{QUB}} = \mathcal{L}(f(x)) + (f(x+\delta) - f(x))^T \nabla_f \mathcal{L}(f(x))$$
$$+ \frac{1}{4} \|f(x+\delta) - f(x)\|_2^2. \quad (7)$$

---

**Algorithm 1** AT with Static QUB Loss

---

**Input:** network architecture $f$ parameterized by $\theta$, batch size $B$, batched training data $\{x_i, y_i\}_{i=1}^{B}$, training epoch $T$, perturbation generation method $P$
**Output:** Adversarially robust network $f$
**for** $t = 1$ **to** $T$ **do**
    **for** $i = 1$ **to** $B$ **do**
        $\delta = P(f, x_i, y_i)$
        Use Equation (7) to compute $\mathcal{L}_{\text{QUB}}$
        $\theta \leftarrow \theta - \nabla_\theta \mathcal{L}_{\text{QUB}}$
    **end for**
**end for**

---

**Algorithm 2** AT w/ Decreasing Weight on QUB Loss

---

**Input:** network architecture $f$ parameterized by $\theta$, batch size $B$, batched training data $\{x_i, y_i\}_{i=1}^{B}$, training epoch $T$, perturbation generation method $P$
**Output:** Adversarially robust network $f$
**for** $t = 1$ **to** $T$ **do**
    $\lambda_t = t/T$
    **for** $i = 1$ **to** $B$ **do**
        $\delta = P(f, x_i, y_i)$
        $\mathcal{L}_{\text{AT}} = \mathcal{L}(f(x_i + \delta), y)$
        Use Equation (7) to compute $\mathcal{L}_{\text{QUB}}$
        $\mathcal{L}_{\text{total}} = (1 - \lambda_t) \cdot \mathcal{L}_{\text{QUB}} + \lambda_t \cdot \mathcal{L}_{\text{AT}}$
        $\theta \leftarrow \theta - \nabla_\theta \mathcal{L}_{\text{total}}$
    **end for**
**end for**

---

## 3.2. Interpretation of QUB Loss

We analyze each term in the QUB loss (7) and discuss how the loss can help improve the robustness against adversarial attacks. The first term, $\mathcal{L}(f(x))$, represents the loss on clean samples, reflecting the model's ability to handle unperturbed data. This term drives the model towards higher standard accuracy (SA), which measures the model's performance on unperturbed data.

The second term, $(f(x + \delta) - f(x))^T \nabla_f \mathcal{L}(f(x))$, can be approximated by applying the chain rule as

$$(f(x+\delta) - f(x))^T \nabla_f \mathcal{L}(f(x)) \approx \delta^T \nabla_x \mathcal{L}(f(x)). \quad (8)$$

The detailed derivation is available in Appendix D. The right-hand side is the inner product between the perturbation $\delta$ and the gradient of the loss with respect to the input. This value is small when the direction of the perturbation does not align with the direction that increases the loss the most. Consequently, minimizing the second term can potentially mitigate the adversarial impact of perturbation on the loss, thereby enhancing robustness. This is closely related to the flatness of the loss landscape: when the loss landscape is flatter (i.e., small $\|\nabla_x \mathcal{L}(f(x))\|$), small perturbations have less impact on the model's performance, leading to

greater robustness (Yu et al., 2018; Li & Spratling, 2023). The second term is the increment of loss by perturbation, and hence, minimizing the second term has the effect of flattening the loss landscape as well.

The third term, $||f(x + \delta) - f(x)||_2^2$, represents the change in the model's logit when perturbation is applied. Obviously, minimizing this term helps limit the impact of adversarial examples on the model's output, further strengthening robustness, as shown in previous works (Zhang et al., 2019; Sriramanan et al., 2021; 2020).

The second and third terms seem to have similar effect in enhancing robustness, however, using both of the terms may achieve better robustness. Specifically, for the same value of the third term, there may be various logit change vector $f(x+\delta) - f(x)$ aligning differently with the gradient $\nabla_x \mathcal{L}(f(x))$. Hence, the second term can further enhance robustness by adjusting the vector $f(x + \delta) - f(x)$ away from the gradient so as to prevent the increase of loss.

**Remark.** One could directly use the right hand side of (8) in order to minimize the change in loss $\mathcal{L}$ when perturbation $\delta$ is applied to input. However, the second term of QUB offers several advantages over the right-hand side, $\delta^T \nabla_x \mathcal{L}(f(x))$. First, as shown in (C.13), the gradient $\nabla_f \mathcal{L}$ can be computed in closed form as the difference between the softmax vector $\hat{y}$ and the one-hot vector $y$, without requiring additional backpropagation steps. Second, the memory usage is also saved. In contrast to the second term, which uses $f(x)$, $f(x + \delta)$, and $\nabla_f \mathcal{L}(f(x))$—all of which lie in $\mathbb{R}^C$, where $C$ is the number of classes—the terms $\delta$ and $\nabla_x \mathcal{L}(f(x))$ require the storage of input-sized tensors, which lie in $\mathbb{R}^{c \times H \times W}$ (or $\mathbb{R}^{c \cdot H \cdot W}$ if vectorized), where $c$ is the number of channels, $H$ is the height, and $W$ is the width of the input. Therefore, using the second term in QUB instead of right hand side in (8) is efficient in both computation and memory usage.

### 3.3. Training Strategy

We propose a generalized training framework based on the QUB loss, as detailed in Algorithm 1. As shown in Eq. (4), we retain the original inner maximization procedure $P$, which corresponds to existing perturbation generation methods and is not the focus of our contribution. Since the QUB loss serves as an upper bound on the standard adversarial training (AT) loss, minimizing the QUB loss has the effect of reducing the AT loss. Therefore, the AT loss can be replaced with the QUB loss in the original outer minimization problem in (4), enabling a standalone adversarial training method.

While this upper-bound property can be beneficial, particularly in the early phase of training where minimizing the QUB loss can quickly improve robustness, it also introduces

a potential drawback. Due to its worst-case nature, the QUB loss tends to produce gradients with larger magnitude compared to the original AT loss. In stochastic gradient descent (SGD), this corresponds to treating the current model as overly pessimistic and applying stronger corrective forces toward robustness. As training progresses and the model becomes sufficiently robust, continuing to rely solely on the QUB loss can lead to excessive regularization, which in turn may degrade standard accuracy. In other words, the model may overcompensate for robustness, sacrificing generalization on clean inputs.

To mitigate this issue, we propose a training strategy in which the model is trained with the QUB loss in the initial phase, and the QUB loss is gradually transitioned to the AT loss as the training progresses. This approach, which we call **QUB-decreasing**, is designed to emphasize robustness in the early phase and progressively restore balance between robustness and generalization. We implement this transition using a simple linear schedule with respect to training epochs, which requires no additional tuning or computational overhead. This choice reflects our goal of maintaining training efficiency and simplicity. This AT method is detailed in Algorithm 2.

## 4. Experiments

### 4.1. Experimental Settings

We conduct experiments to compare the performance of models trained with QUB loss to existing methods adopting the traditional AT loss in (4). We use a single NVIDIA GeForce RTX 4090 GPU with 24GB of memory.

**Datasets and Training Settings.** To evaluate robustness in image classification, we use three datasets: CIFAR-10, CIFAR-100 (Krizhevsky et al., 2009), and Tiny ImageNet (Netzer et al., 2011). Following common settings in adversarial training, we use ResNet18 (He et al., 2016b) and WideResNet34-10 (Zagoruyko & Komodakis, 2016) as backbones for CIFAR-10 and CIFAR-100, and PreActResNet18 (He et al., 2016a) for Tiny ImageNet. The optimizer is SGD (Ruder, 2016), with a learning rate of 0.1, weight decay of 5e-4, and momentum of 0.9. The batch size is set to 128. Training is conducted over 100 epochs, and we utilize a multistep learning rate scheduler that scales the learning rate by 0.1 at epochs 70 and 85.

**Adversarial Attack Settings.** We set the attack budget $\epsilon$ to 8/255 and use a step size $\alpha$ of 2/255 for multi-step attacks such as PGD and TRADES. Each method's hyperparameters are chosen based on the values recommended in the respective papers. We apply early stopping and save the model with the highest robust accuracy using PGD-10 on the validation set, following (Rice et al., 2020).

Table 1. Test robustness (%) on the CIFAR-10 dataset using ResNet18 architecture. Number in bold indicates the best.

| Method | Step | SA | PGD10 | PGD20 | PGD50-10 | AA | Time (h) |
|---|---|---|---|---|---|---|---|
| no AT | - | **94.64** | 0.00 | 0.00 | 0.00 | 0.00 | 0.57 |
| NuAT | 1 | 82.99 | 51.40 | 50.33 | 49.60 | 47.70 | 1.36 |
| GAT | 1 | 81.64 | **54.78** | **53.87** | 53.30 | 47.96 | 1.45 |
| TRADES | 10 | 82.11 | 54.25 | 53.39 | 52.77 | **50.16** | 3.50 |
| Free-AT | 1 | 75.99 | 45.32 | 44.74 | 44.27 | 41.38 | 0.3 |
| + QUB-static | 1 | 72.98 | 46.72 | 46.19 | 45.89 | 42.82 | 0.56 |
| + QUB-decreasing | 1 | 76.10 | 45.58 | 44.89 | 44.35 | 41.60 | 0.56 |
| FGSM-RS | 1 | 84.32 | 47.28 | 45.60 | 44.66 | 43.34 | 0.86 |
| + QUB-static | 1 | 71.13 | 42.96 | 42.19 | 41.54 | 38.48 | 1.16 |
| + QUB-decreasing | 1 | 72.90 | 43.85 | 42.96 | 42.52 | 39.31 | 1.16 |
| FGSM-CKPT | 1 | 90.02 | 41.19 | 38.81 | 37.42 | 37.22 | 1.05 |
| + QUB-static | 1 | 87.63 | 45.41 | 43.78 | 42.54 | 41.53 | 1.35 |
| + QUB-decreasing | 1 | 88.56 | 43.87 | 41.88 | 40.70 | 39.85 | 1.35 |
| FGSM-GA | 1 | 82.93 | 49.89 | 48.53 | 47.74 | 45.75 | 3.02 |
| + QUB-static | 1 | 79.75 | 52.24 | 51.33 | 50.82 | 47.33 | 3.27 |
| + QUB-decreasing | 1 | 81.83 | 50.88 | 49.83 | 49.07 | 46.74 | 3.27 |
| FGSM-PGI(MEP) | 1 | 81.48 | 53.43 | 52.47 | 51.75 | 48.41 | 0.89 |
| + QUB-static | 1 | 80.45 | 53.99 | 53.16 | 52.43 | 48.35 | 1.19 |
| + QUB-decreasing | 1 | 81.56 | 53.95 | 52.99 | 52.24 | 48.58 | 1.19 |
| N-FGSM | 1 | 81.21 | 49.12 | 48.02 | 47.36 | 45.17 | 0.58 |
| + QUB-static | 1 | 80.76 | 51.19 | 50.24 | 49.60 | 47.00 | 0.70 |
| + QUB-decreasing | 1 | 80.77 | 50.30 | 49.35 | 48.70 | 46.60 | 0.70 |
| FGSM-UAP | 1 | 81.62 | 53.38 | 52.59 | 51.83 | 47.75 | 1.18 |
| + QUB-static | 1 | 79.70 | 54.25 | 53.51 | 52.77 | 47.76 | 1.49 |
| + QUB-decreasing | 1 | 80.54 | 54.07 | 53.32 | 52.43 | 47.80 | 1.49 |
| ELLE-A | 1 | 82.14 | 47.91 | 46.39 | 45.57 | 43.52 | 0.97 |
| + QUB-static | 1 | 77.60 | 50.20 | 49.44 | 48.86 | 45.51 | 1.21 |
| + QUB-decreasing | 1 | 80.96 | 49.70 | 48.62 | 47.88 | 45.55 | 1.21 |
| PGD-AT | 10 | 81.53 | 52.99 | 52.30 | 51.82 | 48.33 | 2.34 |
| + QUB-static | 10 | 80.24 | 54.58 | **53.87** | **53.39** | 49.91 | 2.64 |
| + QUB-decreasing | 10 | 82.78 | 53.33 | 52.31 | 51.58 | 49.02 | 2.64 |

**Baselines.** We compare the performance of widely used AT methods including Free-AT (Shafahi et al., 2019), FGSM-RS (Wong et al., 2020), FGSM-GA (Andriushchenko & Flammarion, 2020), FGSM-CKPT (Kim et al., 2021), FGSM-PGI (Jia et al., 2022a), N-FGSM (de Jorge Aranda et al., 2022), FGSM-UAP (Pan et al., 2024), ELLE-A (Rocamora et al., 2024), and PGD-AT (Madry et al., 2018), with and without QUB loss. Methods that avoid applying cross-entropy loss directly to adversarial inputs, such as TRADES (Zhang et al., 2019), NuAT (Sriramanan et al., 2021), and GAT (Sriramanan et al., 2020), are also included purely for performance comparisons without QUB loss.

## 4.2. Impact of QUB on Accuracy

To evaluate the effectiveness of our method, we conduct extensive experiments focusing on accuracy under different adversarial conditions. Using an attack budget of $\epsilon = 8/255$, we assess Standard Accuracy (SA) for original images and Robust Accuracy (RA) against various attacks.

The attacks include:

- **PGD10** and **PGD20** that perform 10 and 20 iterations of perturbation updates, respectively.

- **PGD50-10** (Wong et al., 2020) that applies 50 iterations per restart over 10 restarts, generating significantly stronger perturbations.

- **AutoAttack (AA)** (Croce & Hein, 2020a) that com-

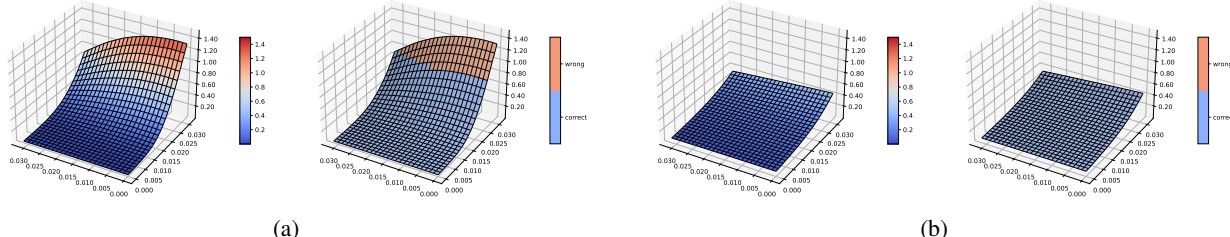

(a)                                                (b)

*Figure 1.* Loss landscape for a specific sample: (a) model trained with FGSM-CKPT and (b) with FGSM-CKPT + QUB. The left side shows colors based on the loss value, and the right side shows colors based on prediction accuracy.

bines multiple attacks—APGD-CE and APGD-DLR with auto-adjusting step size (Croce & Hein, 2020a), Square Attack (Andriushchenko et al., 2020), and FAB attack (Croce & Hein, 2020b) —that provides a comprehensive evaluation of robustness.

Table 1 presents the results on CIFAR-10 using ResNet18 as the backbone architecture. Additional experimental results with different model architectures and datasets are provided in Appendix F.

QUB-static refers to the case where the QUB loss is used throughout the entire training process following Algorithm 1, while QUB-decreasing refers to the approach outlined in Algorithm 2, where the QUB loss is initially used in the early stages of training, gradually transitioning to AT loss as the training progresses.

Interestingly, we observe that combining QUB with FGSM-RS not only fails to improve robustness, but can even degrade performance. This is because FGSM-RS, which is well known to suffer from limited exploration and is highly prone to catastrophic overfitting (Kang & Moosavi-Dezfooli, 2021), often generates suboptimal or misleading adversarial examples. When QUB regularizes the loss landscape around such low-quality perturbations, it may inadvertently enforce smoothness in non-informative or misleading regions, further harming generalization. As a result, QUB may amplify the limitations of FGSM-RS rather than compensating for them.

Except for FGSM-RS, all baselines show an improvement in robustness when applying the QUB loss. This result verifies our discussion in Section 3.2 that the QUB loss can possibly drive the model toward the point where the change in loss is somewhat dampened in the face of perturbations or attacks.

Note that QUB-static achieves better RA than QUB-decreasing for most of AT methods and attacks, while SA is more compromised with QUB-static than with QUB-decreasing. In many cases—such as with Free-AT—the drop in standard accuracy outweighs the gain in robustness accuracy. This result supports our hypothesis in Section 3.3 that QUB, due to its upper-bound nature, may apply stronger

gradients and lead to overemphasis on robustness at the cost of clean performance.

In contrast, QUB-decreasing balances this trade-off more effectively by gradually shifting from QUB loss to AT loss during training. As a result, it achieves comparable or improved RA and better SA in many settings (see Table 1).

In terms of training time, we observe a slight increase due to the additional computation required for each term in the QUB loss. However, compared to multi-step attacks used in methods such as PGD-AT and TRADES, FAT methods with QUB loss still require comparatively less time. Therefore, QUB loss can be considered an effective auxiliary approach in FAT, as it significantly enhances performance with only a modest increase in training time.

### 4.3. Flatness of the Loss Landscape

We visualize the loss landscape on the CIFAR-10 dataset using the ResNet18 architecture.

**Visualization of Loss Landscape**

Visualizing the cross-entropy loss landscape around the input can help assess a model's vulnerability to adversarial attacks. If sharp peaks exist in the loss landscape, it indicates that even a small perturbation can potentially mislead the model to make an incorrect classification. In contrast, a flatter landscape implies greater robustness, as successful attacks may need stronger perturbations of input (Yu et al., 2018; Li & Spratling, 2023).

To create a 3D visualization of the loss landscape, we project a clean image's cross-entropy loss in two directions: the gradient direction ($d_g$) corresponding to the cross-entropy loss gradient and a random direction ($d_r$). We generate a 50×50 grid from 0 to the attack budget $\epsilon$ (8/255) for each direction and compute the loss value at each grid point. This visualization offers insights into the sharpness or flatness of the landscape around the input, showing how resilient the model may be to adversarial perturbations (Chan et al., 2020; Dong et al., 2020; Kim et al., 2021).

Figure 1 shows that models trained with QUB loss exhibit

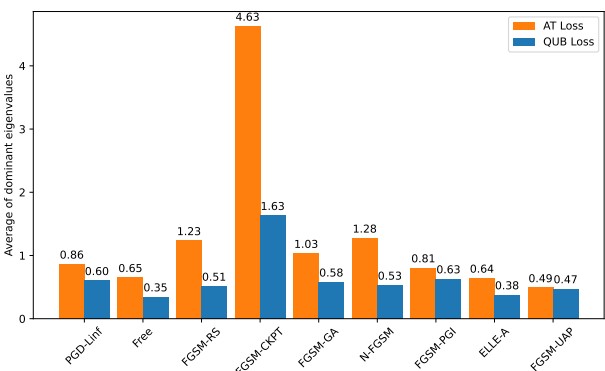

*Figure 2.* Average dominant eigenvalue for each method. Models trained with QUB loss show smaller dominant eigenvalues.

a significantly flatter loss landscape. As mentioned above, this flatter landscape indicates that the model can make accurate predictions across a broader region around each input sample. Consequently, the model's robustness is enhanced, as it becomes less sensitive to small perturbations around the input. This result demonstrates that our QUB loss effectively strengthens robustness and more importantly, trains the model to remain robust against unseen attacks.

**Dominant Eigenvalue of Hessian Matrix**

While the visualization verifies the flatness of loss landscape around a single sample, it gives only a limited view of landscape over the entire dataset. To further assess the flatness of the loss landscape, we examine the dominant eigenvalue of the Hessian matrix of the cross-entropy loss with respect to the input. The eigenvalue of the Hessian matrix reflects the curvature along specific directions; a larger eigenvalue suggests a steeper curvature along the corresponding eigenvector. By focusing on the dominant (largest) eigenvalue, we can capture the overall sharpness or flatness of the loss landscape. A smaller dominant eigenvalue indicates a flatter landscape, which is typically associated with greater robustness.

In this experiment, we extract 1,000 samples from the CIFAR-10 test dataset and calculate the dominant eigenvalue of the Hessian matrix of the cross-entropy loss with respect to each sample. We then compute the average of these dominant eigenvalues to compare the flatness of the loss landscape across different training methods.

As shown in Figure 2, the use of QUB loss results in generally smaller eigenvalues across the dataset. This indicates that the loss landscape is not only flattened for specific samples as shown in Figure 1, but also exhibits enhanced robustness across the entire dataset. This experiment empirically supports the claim made in Section 3.2 that the second term in QUB loss contributes to flattening the loss landscape. Moreover, the upper bound loss helps the model

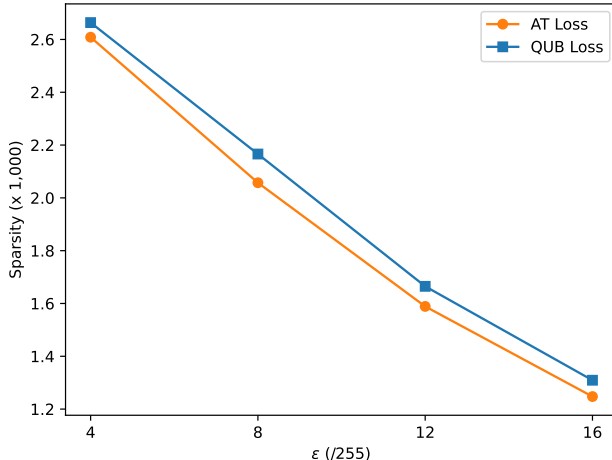

*Figure 3.* Comparing FGSM-CKPT (blue line) and FGSM-CKPT + QUB (orange line) for each attack budget shows that the sparsity value with QUB is consistently higher in all ranges.

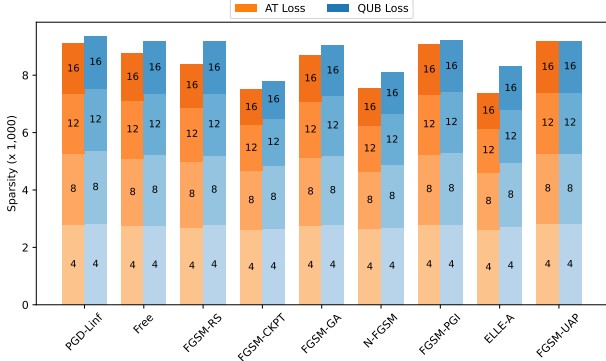

*Figure 4.* Sparsity values with and without QUB for each method. Using QUB consistently results in higher values across all methods.

defend not only against the perturbations used in training but also against potential attacks that are unseen in training, suggesting robustness against broader adversarial scenarios.

**4.4. Adversarial Sparsity**

Several studies suggest that RA alone cannot fully capture the robustness of a model, as it focuses solely on whether an attack point is defended. The sparsity metric (Olivier & Raj, 2023) addresses this by measuring the average distance to the nearest attackable point within $L_\infty$ ball, with a larger value indicating greater robustness. The experiments are conducted in the same environment as in Section 4.3, and we vary the attack budget $\epsilon$ across [4/255, 8/255, 12/255, 16/255] as opposed to the training with fixed $\epsilon$ of 8/255.

As shown in Figure 3, applying QUB loss during training on FGSM-CKPT results in better performance in terms of the density metric of attackable points compared to training with AT loss on FGSM-CKPT. Figure 4 illustrates the sparsity of

each AT method measured at four different values of epsilon. In all the methods, the sparsity values are consistently larger for every attack budget, indicating that the QUB loss helps reduce the number of attackable points.

## 5. Conclusion

In this paper, we presented a novel FAT method for enhancing the robustness of a model against adversarial attacks. Our method aims to drive the model toward minimizing a quadratic upper bound on the cross-entropy loss function, instead of the conventional AT loss function. Our method can be applied to many existing FAT methods that are based on the conventional AT loss function. We showed through extensive experiments that the robustness of existing FAT methods can be further enhanced with the QUB loss. Various metrics such as eigenvalues and sparsity demonstrate that the QUB loss has the effect of flattening the loss landscape, which contributes to enhanced robustness.

## Acknowledgments

This work was supported by the National Research Foundation of Korea(NRF) grant funded by the Korea government(MSIT)(RS-2025-00516578). This paper was written as part of Konkuk University's research support program for its faculty on sabbatical leave in 2024.

## Impact Statement

We develop a novel adversarial training method to enhance the robustness of machine learning models against adversarial attacks by incorporating a new loss function into existing training approaches. As a result, the proposed method significantly improves the resilience of models, making them more reliable in real-world applications, especially in critical areas such as autonomous systems, healthcare and finance. This advancement is important because it helps secure AI systems against potential malicious attacks, fostering greater trust in AI technologies and ensuring their safe, ethical deployment in high-stakes environments.

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

## A. Proof of convexity of loss function

Although the convexity of cross-entropy loss function with respect to the logit is known in the literature, we present the proof for the completeness of the paper. A function $g$ is convex if it satisfies the following inequality for any two vectors $a$ and $b$ in the domain of $g$:

$$g(\lambda a + (1 - \lambda)b) \leq \lambda g(a) + (1 - \lambda)g(b), \quad \forall \lambda \in [0, 1]. \tag{A.1}$$

We will use this definition to prove the convexity of the loss function with respect to the logit.

The cross-entropy loss is a nonnegative linear combination of functions of the following form:

$$g(z) = -\log\left(\frac{e^{z_i}}{\sum_{k=1}^{C} e^{z_k}}\right), \tag{A.2}$$

where $z_i$ is the $i$-th element of the logit vector, and $C$ is the number of classes. Since a nonnegative combination of convex functions is convex, it suffices to prove the convexity of $g(z)$. The function $g(z)$ is written as:

$$g(z) = -z_i + \log\left(\sum_{k=1}^{C} e^{z_k}\right). \tag{A.3}$$

Since the linear function is convex, we just need to prove the convexity of the second term. We invoke Hölder's inequality for the proof.

Consider $p, q \in [1, \infty]$ such that $\frac{1}{p} + \frac{1}{q} = 1$. Hölder's inequality states that:

$$\sum_{i=1}^{n} u_i v_i \leq \left(\sum_{i=1}^{n} |u_i|^p\right)^{\frac{1}{p}} \cdot \left(\sum_{i=1}^{n} |v_i|^q\right)^{\frac{1}{q}}, \tag{A.4}$$

for any two vectors $u = (u_1, \ldots, u_n)$ and $v = (v_1, \ldots, v_n)$ of real numbers. To apply this inequality, let $u_i = e^{\lambda a_i}$ and $v_i = e^{(1-\lambda)b_i}$, where $a_i$'s and $b_i$'s are arbitrary real numbers and $\lambda$ is a constant that lies within the interval $(0, 1)$. Choose $p = \frac{1}{\lambda}$ and $q = \frac{1}{1-\lambda}$, which satisfies $\frac{1}{p} + \frac{1}{q} = 1$. Hölder's inequality gives:

$$\sum_{i=1}^{C} e^{\lambda a_i} e^{(1-\lambda)b_i} \leq \left(\sum_{i=1}^{C} e^{a_i}\right)^{\lambda} \cdot \left(\sum_{i=1}^{C} e^{b_i}\right)^{1-\lambda}. \tag{A.5}$$

Taking the logarithm on both sides yields:

$$\log\left(\sum_{i=1}^{C} e^{\lambda a_i + (1-\lambda)b_i}\right) \leq \lambda \log\left(\sum_{i=1}^{C} e^{a_i}\right) + (1 - \lambda)\log\left(\sum_{i=1}^{C} e^{b_i}\right). \tag{A.6}$$

This implies that the second term in (A.3) is convex, and consequently, $g(z)$ is convex. This completes the proof.

## B. Proof of Lemma 1

We derive a quadratic upper bound for convex functions using the two well-known facts in the following.

**Lemma 3** (Taylor's Theorem, (Apostol, 1975)). *For any twice continuously differentiable function $g$,*

$$g(y) = g(x) + \nabla g(x)^T (y - x) + \frac{1}{2}(y - x)^T \nabla^2 g(z)(y - x), \quad \forall x, y, \tag{B.1}$$

*where $z$ is some vector between $x$ and $y$ (i.e., $z = \alpha x + (1 - \alpha)y$ for some $\alpha \in [0, 1]$).*

Note that $\nabla^2 g(z)$ denotes the Hessian of $g$, which represents the matrix of second-order partial derivatives of the function $g$ evaluated at the point $z$. This Hessian matrix is denoted as $\boldsymbol{H}$ in the following sections.

**Lemma 4** ((Horn & Johnson, 1985)). *For a symmetric matrix A,*

$$v^T \boldsymbol{A} v \leq \lambda_{\max}(\boldsymbol{A}) \cdot \|v\|^2, \tag{B.2}$$

*where $\lambda_{\max}(\boldsymbol{A})$ is the largest eigenvalue of $\boldsymbol{A}$.*

By utilizing the symmetry of the Hessian matrix, and applying inequality (B.2) to equation (B.1), we derive the following inequality:

$$g(y) \leq g(x) + \nabla g(x)^T (y - x) + \frac{\lambda_{\max}(\boldsymbol{H})}{2} \|y - x\|^2. \tag{B.3}$$

For a convex function $g$, its Hessian $\boldsymbol{H}$ is always positive semidefinite and hence, the eigenvalues of $\boldsymbol{H}$ are all nonnegative. Furthermore, the largest eigenvalue of $\boldsymbol{H}$ can be computed as follows:

$$\|\boldsymbol{H}\|_2 = \max_{\|v\|_2=1} \|\boldsymbol{H}v\|_2 \tag{B.4}$$

$$= \max_{\|v\|_2=1} \sqrt{v^T \boldsymbol{H}^T \boldsymbol{H} v} \tag{B.5}$$

$$= \max_{\|v\|_2=1} \sqrt{v^T \boldsymbol{H}^2 v} \qquad (\because \boldsymbol{H} \text{ is symmetric}) \tag{B.6}$$

$$= \sqrt{\lambda_{\max}(\boldsymbol{H}^2)} \tag{B.7}$$

$$= \sqrt{\lambda_{\max}^2(\boldsymbol{H})} \tag{B.8}$$

$$= \lambda_{\max}(\boldsymbol{H}). \qquad (\because g \text{ is convex} \equiv \boldsymbol{H} \succeq 0) \tag{B.9}$$

Therefore, we have the following quadratic upper bound on the convex function $g$:

$$g(y) \leq g(x) + \nabla g(x)^T (y - x) + \frac{\|\boldsymbol{H}\|_2}{2} \|y - x\|^2. \tag{B.10}$$

This together with the convexity of $\mathcal{L}(f(x))$ with respect to the logic $f(x)$ (Appendix A) yields

$$\mathcal{L}(f(x + \delta)) \leq \mathcal{L}(f(x)) + (f(x + \delta) - f(x))^T \nabla_f \mathcal{L}(f(x))$$
$$+ \frac{\|\boldsymbol{H}\|_2}{2} \|f(x + \delta) - f(x)\|_2^2, \tag{B.11}$$

where $\boldsymbol{H}$ is the Hessian of the loss with respect to logits, evaluated at some point between $f(x)$ and $f(x + \delta)$, i.e., $\boldsymbol{H} = \nabla_f^2 \mathcal{L}(\tilde{f})$ with $\tilde{f} = \alpha f(x) + (1 - \alpha) f(x + \delta)$ for some $\alpha \in [0, 1]$. This completes the proof.

**Remark.** An alternative derivation of the QUB loss can be made using the notion of $\beta$-smoothness, a standard tool in optimization theory. In particular, one may derive the bound by first assuming that the cross-entropy loss has a Hessian with bounded $l_2$-norm and then applying the lemma that such a condition implies $\beta$-Lipschitz continuity of the gradient. This yields the same upper bound as in (B.1), and offers a more succinct interpretation (see, e.g., (Hazan, 2019)). We chose to present the derivation via Taylor's theorem to explicitly connect the upper bound to the local behavior of the loss landscape.

## C. Proof of Lemma 2

### Derivative of Softmax function

For the convenience of the notation, we will use $z$ to represent the logits instead of $f(x)$. The probability of each class after applying the softmax function is given by:

$$\hat{y}_i = \frac{e^{z_i}}{\sum_{j=1}^C e^{z_j}}, \tag{C.1}$$

where $z_i$ is the $i$-th value of the logit vector $z$. The derivative of $\hat{y}_i$ with respect to the logit $z_n$ is written as follows. When the indices are the same (i.e., $i = n$),

$$\frac{\partial \hat{y}_i}{\partial z_i} = \frac{e^{z_i} \sum_{j=1}^C e^{z_j} - (e^{z_i})^2}{\left(\sum_{j=1}^C e^{z_j}\right)^2} = \frac{e^{z_i}}{\sum_{j=1}^C e^{z_j}} \cdot \frac{\sum_{j=1}^C e^{z_j} - e^{z_i}}{\sum_{j=1}^C e^{z_j}} = \hat{y}_i(1 - \hat{y}_i). \tag{C.2}$$

When the indices are different (i.e., $i \neq n$),

$$\frac{\partial \hat{y}_i}{\partial z_n} = \frac{0 - e^{z_i} e^{z_n}}{\left(\sum_{j=1}^{C} e^{z_j}\right)^2} = -\hat{y}_i \hat{y}_n. \tag{C.3}$$

Combining both cases, we have

$$\frac{\partial \hat{y}_i}{\partial z_n} = \begin{cases} \hat{y}_i(1 - \hat{y}_i), & \text{if } i = n \\ -\hat{y}_i \hat{y}_n, & \text{if } i \neq n \end{cases} \tag{C.4}$$

**Derivative of Cross-Entropy Loss**

Let $y_i$ be the $i$-th element of the one-hot encoding vector $y$, where $y_i = 1$ for the correct class and $y_i = 0$ otherwise. The cross-entropy loss function is defined as

$$\mathcal{L}(z) = -\sum_{i=1}^{C} y_i \log(\hat{y}_i). \tag{C.5}$$

Now, the derivative of the loss with respect to $z_n$ is

$$\frac{\partial \mathcal{L}(z)}{\partial z_n} = -\sum_{i=1}^{C} y_i \frac{\partial \log(\hat{y}_i)}{\partial z_n} \tag{C.6}$$

$$= -\sum_{i=1}^{C} y_i \frac{\partial \log(\hat{y}_i)}{\partial \hat{y}_i} \frac{\partial \hat{y}_i}{\partial z_n} \tag{C.7}$$

$$= -\sum_{i=1}^{C} \frac{y_i}{\hat{y}_i} \frac{\partial \hat{y}_i}{\partial z_n} \tag{C.8}$$

$$= -\frac{y_n}{\hat{y}_n} \hat{y}_n(1 - \hat{y}_n) + \sum_{i \neq n} \frac{y_i}{\hat{y}_i} \hat{y}_i \hat{y}_n \tag{C.9}$$

$$= -y_n + y_n \hat{y}_n + \sum_{i \neq n} y_i \hat{y}_n \tag{C.10}$$

$$= -y_n + \sum_{i=1}^{c} y_i \hat{y}_n \tag{C.11}$$

$$= -y_n + \hat{y}_n. \tag{C.12}$$

Thus, we have

$$\frac{\partial \mathcal{L}(z)}{\partial z_n} = -y_n + \hat{y}_n \quad \text{and} \quad \frac{\partial \mathcal{L}(z)}{\partial z} = -y + \hat{y}. \tag{C.13}$$

Therefore, the derivative of the loss with respect to the logits is simply the difference between the softmax vector $\hat{y}$ and the one-hot encoded vector $y$.

**Second Derivative (Hessian)**

The second derivative of the loss is computed as follows:

$$\frac{\partial^2 \mathcal{L}(z)}{\partial z^2} = \frac{\partial(-y + \hat{y})}{\partial z} = \frac{\partial \hat{y}}{\partial z}. \tag{C.14}$$

Since $y$ is a one-hot encoded vector, it does not depend on the logits and is treated as constant. Therefore, we focus on the derivative of $\hat{y}$ with respect to $z$. The Hessian matrix is given by:

$$\boldsymbol{H} = \begin{bmatrix} \hat{y}_1 - \hat{y}_1^2 & -\hat{y}_1\hat{y}_2 & \cdots & -\hat{y}_1\hat{y}_n \\ -\hat{y}_2\hat{y}_1 & \hat{y}_2 - \hat{y}_2^2 & \cdots & -\hat{y}_2\hat{y}_n \\ \vdots & \vdots & \ddots & \vdots \\ -\hat{y}_n\hat{y}_1 & -\hat{y}_n\hat{y}_2 & \cdots & \hat{y}_n - \hat{y}_n^2 \end{bmatrix} \tag{C.15}$$

or equivalently:

$$H_{ij} = \begin{cases} \hat{y}_i - \hat{y}_i^2, & \text{if } i = j \\ -\hat{y}_i\hat{y}_j, & \text{if } i \neq j \end{cases} \tag{C.16}$$

Calculating the $L_2$ norm of the matrix is computationally expensive. Instead, we use the following upper bound for approximation (Bertsekas & Tsitsiklis, 1997):

$$\|\boldsymbol{H}\|_2^2 \leq \|\boldsymbol{H}\|_1 \cdot \|\boldsymbol{H}\|_\infty, \tag{C.17}$$

where $\|\mathrm{H}\|_1$ and $\|\mathrm{H}\|_\infty$ are the $L_1$ and $L_\infty$ norms of the Hessian matrix, representing the maximum absolute column sum and row sum, respectively. Using the properties of the softmax vector, where $0 < \hat{y}_i < 1$ and $\sum_i \hat{y}_i = 1$, we can calculate the $L_1$ norm of the Hessian matrix:

$$\|\boldsymbol{H}\|_1 = \max_{1 \leq j \leq m} \sum_{i=1}^n |H_{ij}| \tag{C.18}$$

$$= \max_{1 \leq j \leq m} \left( \sum_{i=1}^n \hat{y}_j\hat{y}_i + \hat{y}_j - 2\hat{y}_j^2 \right) \tag{C.19}$$

$$= \max_{1 \leq j \leq m} (2\hat{y}_j - 2\hat{y}_j^2). \tag{C.20}$$

The maximum value of this expression occurs when $\hat{y}_j = \frac{1}{2}$, yielding $\frac{1}{2}$. Since the Hessian matrix is symmetric, its $L_\infty$ norm is also $\frac{1}{2}$. Therefore, we have:

$$\|\boldsymbol{H}\|_2^2 \leq \frac{1}{2} \cdot \frac{1}{2} = \frac{1}{4}. \tag{C.21}$$

This completes the proof.

## D. Expressing the Second Term Using Chain Rule

In this section, we further discuss the interpretation of the second term of the QUB loss. Recall that $f_\theta(x)$ represents the logit vector of a neural network model parameterized by $\theta$, when the input is $x$. For convenience, we denote it as $f(x)$. For a sufficiently small $\delta$, the Taylor expansion of $f(x + \delta)$ around $x$ is given by

$$f(x + \delta) \approx f(x) + \nabla_x f(x)\delta, \tag{D.1}$$

where $\nabla_x f(x)$ is the Jacobian matrix of the logit with respect to the input $x$. It lies in $\mathbb{R}^{C \times (c \cdot H \cdot W)}$, where $C$ is the number of classes, $c$ is the number of channels, $H$ is the height, and $W$ is the width of the input. As the notation indicates, we assume that the input vector $x$ or perturbation $\delta$ is vectorized appropriately.

Using the above approximation, we substitute it into the second term of QUB loss, $(f(x + \delta) - f(x))^T \nabla_f \mathcal{L}(f(x))$ as

$$(f(x + \delta) - f(x))^T \nabla_f \mathcal{L}(f(x)) \approx (\nabla_x f(x)\delta)^T \nabla_f \mathcal{L}(f(x)) \tag{D.2}$$

$$= \delta^T \nabla_x f(x)^T \nabla_f \mathcal{L}(f(x)). \tag{D.3}$$

By the chain rule, the gradient of $\mathcal{L}(f(x))$ with respect to $x$ can be written as

$$\nabla_x \mathcal{L}(f(x)) = \nabla_x f(x)^T \nabla_f \mathcal{L}(f(x)). \tag{D.4}$$

It follows that

$$\delta^T \nabla_x f(x)^T \nabla_f \mathcal{L}(f(x)) = \delta^T \nabla_x \mathcal{L}(f(x)). \tag{D.5}$$

We thus have

$$(f(x + \delta) - f(x))^T \nabla_f \mathcal{L}(f(x)) \approx \delta^T \nabla_x \mathcal{L}(f(x)). \tag{D.6}$$

This completes the derivation, showing that the expression $(f(x + \delta) - f(x))^T \nabla_f \mathcal{L}(f(x))$ can be approximated by $\delta^T \nabla_x \mathcal{L}(f(x))$ for small $\delta$.

## E. Performance Comparison of QUB Loss with Respect to Perturbation Budget

The QUB is derived by applying Taylor's Theorem around $f(x)$ with perturbation $\delta$ on $x$, and hence, the bound is effective (i.e., captures the core characteristics of AT loss) when the perturbation $\delta$ is small. We conduct experiments to verify how the QUB loss affects performance across various ranges of attacks, including FGSM-RS, N-FGSM, and PGD-AT. These experiments are performed on the CIFAR-10 dataset using the ResNet-18 architecture.

The training and evaluation are conducted using four different values of $\epsilon$: [4/255, 8/255, 12/255, 16/255]. The rest of the experimental setup is the same as the one used in Section 4.

*Table 2.* Comparison of SA and AA performance under different perturbation budgets for each epsilon. **Bold numbers** indicate the best result within each attack method (i.e., FGSM-RS, N-FGSM, and PGD-AT).

| Perturbation (/255) | 4.0 | | 8.0 | | 12.0 | | 16.0 | |
|---|---|---|---|---|---|---|---|---|
| **Method** | **SA** | **AA** | **SA** | **AA** | **SA** | **AA** | **SA** | **AA** |
| FGSM-RS | **89.46** | 67.42 | **84.32** | 43.34 | **81.48** | 25.79 | **81.21** | 12.94 |
| + QUB-static | 87.92 | **68.52** | 71.13 | 38.48 | 78.11 | **29.68** | 76.21 | 17.06 |
| + QUB-decreasing | 88.47 | 67.74 | 72.90 | **39.31** | 49.22 | 22.66 | 39.46 | **17.15** |
| N-FGSM | **89.32** | 66.64 | **82.57** | 48.98 | **74.18** | 39.45 | **65.43** | 34.97 |
| + QUB-static | 87.66 | **69.33** | 78.96 | 51.32 | 73.84 | **41.34** | 62.86 | 35.84 |
| + QUB-decreasing | 88.42 | 68.43 | 80.92 | **50.69** | 72.16 | 40.42 | 64.39 | **35.92** |
| PGD-AT | **89.04** | 68.03 | 81.19 | 48.35 | **73.35** | 34.81 | 67.69 | 24.48 |
| + QUB-static | 87.90 | **70.09** | 80.58 | **50.07** | 72.71 | **36.16** | 67.38 | 25.18 |
| + QUB-decreasing | 88.24 | 69.32 | **82.78** | 49.47 | 72.57 | 35.89 | **69.60** | **25.20** |

Table 2 presents the results for clean image classification performance (SA) and robustness (AA) across various values of the budget $\epsilon$ on the perturbation $\delta$. To effectively illustrate the changes in values, Figure 5 depicts how the performance metrics evolve relative to the PGD-AT and N-FGSM baselines using AT loss.

During training, when AT loss is replaced with QUB loss, both the standard version (+QUB-static) and its variant with increasing perturbations (+QUB-decreasing) exhibit a gradual decline in robustness as the magnitude of perturbation increases. This is somewhat natural in that the gap between the upper bound and the AT loss increases as the perturbation $\delta$ increases. The QUB loss should thus be used with careful consideration of the attack environment. Nonetheless, this result shows that the QUB can be used to enhance the robustness against some range of attacks. This trend is generally observed in both PGD and N-FGSM, while FGSM-RS does not follow this pattern due to catastrophic overfitting, which distorts the relationship between perturbation size and robustness.

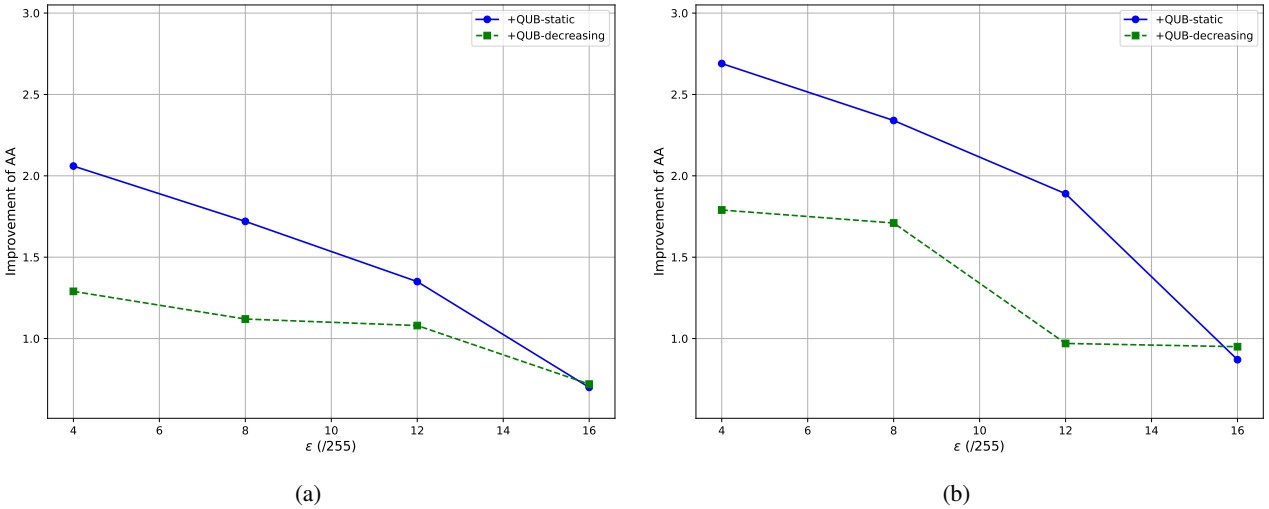

(a)                                    (b)

*Figure 5.* Accuracy improvement from applying QUB to adversarial training using (a) PGD-AT and (b) N-FGSM. While QUB-decreasing shows smaller AA gains compared to QUB-static, it better preserves standard accuracy. As the perturbation strength $\epsilon$ increases, the improvement gradually diminishes, reflecting the characteristics of QUB.

# F. Experiments Results

To verify the effectiveness of our method across various datasets and models, we conduct the experiments presented below (Tables 3, 4, 5, and 6). For CIFAR100 and Tiny ImageNet datasets, we include only QUB-decreasing as QUB-static exhibits excessively deteriorated standard accuracy for those datasets. In all the experiments, we observe that the robustness of many existing AT methods can be enhanced by our QUB loss.

## F.1. CIFAR10, WRN34-10

*Table 3.* Test robustness (%) on the CIFAR-10 dataset using WRN34-10 architecture. Numbers in bold indicate the best.

| Method | Step | SA | PGD10 | PGD20 | PGD50-10 | AA | Time (h) |
|---|---|---|---|---|---|---|---|
| Natural Training | - | **95.07** | 0.00 | 0.00 | 0.00 | 0.00 | 2.67 |
| NuAT | 1 | 85.16 | 53.96 | 52.74 | 51.81 | 49.90 | 9.37 |
| GAT | 1 | 83.44 | **56.17** | 55.39 | **55.00** | 50.65 | 9.85 |
| TRADES | 10 | 84.97 | 55.38 | 54.59 | 54.22 | **51.86** | 40.45 |
| Free-AT | 1 | 80.05 | 42.64 | 41.80 | 41.25 | 39.85 | 2.15 |
| + QUB-static | 1 | 76.42 | 47.06 | 46.51 | 46.08 | 43.59 | 4.29 |
| + QUB-decreasing | 1 | 78.83 | 45.53 | 44.59 | 44.07 | 41.97 | 4.29 |
| FGSM-RS | 1 | 87.05 | 48.70 | 46.93 | 46.03 | 45.18 | 5.02 |
| + QUB-static | 1 | 80.89 | 50.64 | 49.70 | 49.09 | 46.65 | 7.45 |
| + QUB-decreasing | 1 | 69.65 | 40.62 | 39.88 | 39.43 | 36.78 | 7.45 |
| FGSM-CKPT | 1 | 91.45 | 43.87 | 41.31 | 40.10 | 39.49 | 8.68 |
| + QUB-static | 1 | 87.63 | 48.14 | 46.66 | 45.80 | 44.18 | 11.23 |
| + QUB-decreasing | 1 | 91.01 | 46.03 | 43.55 | 42.49 | 42.26 | 11.23 |
| FGSM-GA | 1 | 85.03 | 52.15 | 50.81 | 49.88 | 48.27 | 20.82 |
| + QUB-static | 1 | 82.86 | 54.27 | 53.59 | 52.76 | 50.32 | 22.42 |
| + QUB-decreasing | 1 | 85.05 | 53.47 | 52.35 | 51.66 | 49.74 | 22.42 |
| N-FGSM | 1 | 84.26 | 50.01 | 48.67 | 47.98 | 46.64 | 3.44 |
| + QUB-static | 1 | 85.39 | 53.39 | 52.12 | 51.36 | 49.91 | 5.04 |
| + QUB-decreasing | 1 | 82.66 | 51.31 | 51.29 | 50.66 | 48.86 | 5.04 |
| FGSM-PGI(MEP) | 1 | 80.36 | 52.84 | 52.33 | 51.68 | 48.59 | 5.89 |
| + QUB-static | 1 | 84.66 | 54.26 | 53.23 | 52.33 | 49.14 | 8.32 |
| + QUB-decreasing | 1 | 84.31 | 55.28 | 54.08 | 53.37 | 50.10 | 8.32 |
| ELLE-A | 1 | 80.78 | 46.95 | 45.80 | 45.20 | 43.16 | 8.49 |
| + QUB-static | 1 | 78.97 | 49.93 | 48.86 | 48.42 | 46.92 | 10.04 |
| + QUB-decreasing | 1 | 81.00 | 47.97 | 46.71 | 46.10 | 44.14 | 10.04 |
| FGSM-UAP | 1 | 84.73 | 53.93 | 52.64 | 51.44 | 48.84 | 7.49 |
| + QUB-static | 1 | 84.47 | 54.46 | 53.42 | 52.68 | 49.72 | 9.92 |
| + QUB-decreasing | 1 | 84.54 | 54.31 | 53.12 | 52.29 | 49.52 | 9.92 |
| PGD AT | 10 | 84.88 | 53.99 | 52.98 | 52.46 | 49.80 | 24.24 |
| + QUB-static | 10 | 82.66 | 56.14 | **55.47** | 54.99 | 51.49 | 26.63 |
| + QUB-decreasing | 10 | 85.21 | 55.12 | 54.17 | 53.66 | 50.86 | 26.63 |

## F.2. CIFAR100, ResNet18

*Table 4.* Test robustness (%) on the CIFAR-100 dataset using ResNet18 architecture. Numbers in bold indicate the best.

| Method | Step | SA | PGD10 | PGD20 | PGD50-10 | AA | Time (h) |
|---|---|---|---|---|---|---|---|
| Natural Training | - | **76.92** | 0.02 | 0.02 | 0.00 | 0.00 | 0.92 |
| NuAT | 1 | 54.90 | 25.50 | 22.14 | 19.44 | 18.79 | 3.01 |
| GAT | 1 | 59.92 | 28.37 | 27.76 | 27.09 | 22.87 | 2.81 |
| TRADES | 10 | 57.69 | 30.22 | 29.78 | 29.15 | 25.06 | 7.17 |
| Free-AT | 1 | 48.42 | 23.27 | 22.85 | 22.56 | 19.03 | 0.30 |
| + QUB-decreasing | 1 | 44.36 | 23.85 | 23.61 | 23.49 | 19.71 | 0.56 |
| FGSM-RS | 1 | 45.39 | 19.98 | 19.41 | 18.87 | 15.93 | 1.16 |
| + QUB-decreasing | 1 | 33.57 | 19.86 | 18.62 | 18.48 | 15.16 | 1.60 |
| FGSM-CKPT | 1 | 69.79 | 12.68 | 9.55 | 7.39 | 5.71 | 1.34 |
| + QUB-decreasing | 1 | 63.16 | 24.97 | 23.96 | 23.27 | 21.54 | 2.16 |
| FGSM-GA | 1 | 56.38 | 28.37 | 27.55 | 26.98 | 23.43 | 3.03 |
| + QUB-decreasing | 1 | 51.11 | 29.09 | 29.46 | 29.12 | 25.11 | 3.28 |
| N-FGSM | 1 | 55.40 | 27.18 | 26.69 | 26.25 | 23.36 | 1.21 |
| + QUB-decreasing | 1 | 51.99 | **30.46** | **30.14** | **29.78** | **25.82** | 1.97 |
| FGSM-PGI(MEP) | 1 | 55.69 | 29.21 | 28.55 | 28.23 | 24.48 | 2.83 |
| + QUB-decreasing | 1 | 53.68 | 29.78 | 29.20 | 28.77 | 24.92 | 3.46 |
| ELLE-A | 1 | 54.36 | 25.53 | 24.80 | 23.92 | 21.03 | 1.15 |
| + QUB-decreasing | 1 | 52.13 | 27.39 | 26.81 | 26.39 | 22.93 | 1.30 |
| FGSM-UAP | 1 | 56.57 | 28.83 | 28.23 | 27.66 | 24.39 | 3.70 |
| + QUB-decreasing | 1 | 55.01 | 29.28 | 29.13 | 28.76 | 25.15 | 4.25 |
| PGD AT | 10 | 55.52 | 29.14 | 28.85 | 28.38 | 24.22 | 5.66 |
| + QUB-decreasing | 10 | 53.49 | 29.98 | 29.61 | 29.25 | 25.40 | 6.29 |

## F.3. CIFAR100, WRN34-10

*Table 5.* Test robustness (%) on the CIFAR-100 dataset using WRN34-10 architecture. Numbers in bold indicate the best.

| Method | Step | SA | PGD10 | PGD20 | PGD50-10 | AA | Time (h) |
|---|---|---|---|---|---|---|---|
| Natural Training | - | **78.89** | 0.04 | 0.01 | 0.00 | 0.00 | 4.9 |
| NuAT | 1 | 49.43 | 20.45 | 18.44 | 16.41 | 16.94 | 9.53 |
| GAT | 1 | 65.71 | 25.92 | 25.05 | 24.32 | 22.44 | 9.93 |
| TRADES | 10 | 61.14 | **31.55** | **31.16** | 30.62 | 27.17 | 40.54 |
| Free-AT | 1 | 48.74 | 21.82 | 21.38 | 21.00 | 18.45 | 2.15 |
| + QUB-decreasing | 1 | 44.13 | 24.70 | 24.40 | 23.92 | 21.01 | 4.28 |
| FGSM-RS | 1 | 62.48 | 26.80 | 25.46 | 24.70 | 23.25 | 5.97 |
| + QUB-decreasing | 1 | 38.24 | 19.54 | 19.42 | 19.26 | 15.86 | 7.44 |
| FGSM-CKPT | 1 | 60.66 | 8.89 | 7.25 | 5.31 | 3.16 | 8.21 |
| + QUB-decreasing | 1 | 51.82 | 18.95 | 18.02 | 17.40 | 15.79 | 11.26 |
| FGSM-GA | 1 | 60.44 | 27.48 | 26.45 | 25.78 | 23.92 | 20.70 |
| + QUB-decreasing | 1 | 53.60 | 30.16 | 29.64 | 29.13 | 25.85 | 23.87 |
| N-FGSM | 1 | 58.85 | 29.08 | 28.36 | 27.84 | 25.34 | 3.48 |
| + QUB-decreasing | 1 | 53.10 | 30.88 | 30.49 | 30.15 | 26.41 | 5.04 |
| FGSM-PGI(MEP) | 1 | 61.95 | 30.30 | 29.26 | 28.20 | 25.80 | 5.86 |
| + QUB-decreasing | 1 | 59.22 | 30.38 | 29.43 | 28.77 | 26.20 | 8.29 |
| ELLE-A | 1 | 55.24 | 29.63 | 29.71 | 29.28 | 22.92 | 8.50 |
| + QUB-decreasing | 1 | 54.02 | 26.43 | 26.12 | 25.48 | 18.64 | 9.96 |
| FGSM-UAP | 1 | 61.00 | 29.49 | 28.55 | 27.68 | 24.87 | 7.52 |
| + QUB-decreasing | 1 | 59.19 | 29.97 | 29.39 | 28.85 | 26.03 | 9.91 |
| PGD AT | 10 | 58.56 | 29.95 | 29.37 | 28.98 | 25.70 | 24.20 |
| + QUB-decreasing | 10 | 52.98 | 31.41 | 31.11 | **30.85** | **27.20** | 26.63 |

## F.4. Tiny ImageNet, PreActResNet18

*Table 6.* Test robustness (%) on the Tiny ImageNet dataset using PreActResNet18 architecture. Numbers in bold indicate the best.

| Method | Step | SA | PGD10 | PGD20 | PGD50-10 | AA | Time (h) |
|---|---|---|---|---|---|---|---|
| Natural Training | - | **63.27** | 0.01 | 0.01 | 0.00 | 0.00 | 2.06 |
| NuAT | 1 | 53.97 | 18.25 | 17.52 | 17.06 | 13.50 | 5.72 |
| GAT | 1 | 49.75 | 17.71 | 17.30 | 16.93 | 12.62 | 6.11 |
| TRADES | 10 | 46.70 | 21.76 | 21.56 | 21.38 | 16.33 | 16.55 |
| Free-AT | 1 | 36.32 | 16.22 | 16.04 | 15.91 | 12.03 | 1.29 |
| + QUB-decreasing | 1 | 31.34 | 16.59 | 16.47 | 16.38 | 12.51 | 2.52 |
| FGSM-RS | 1 | 48.14 | 20.54 | 19.98 | 19.67 | 16.39 | 3.39 |
| + QUB-decreasing | 1 | 29.47 | 16.02 | 15.89 | 15.77 | 11.73 | 5.10 |
| FGSM-CKPT | 1 | 57.93 | 11.68 | 10.49 | 9.89 | 9.07 | 4.16 |
| + QUB-decreasing | 1 | 52.11 | 19.13 | 18.33 | 17.83 | 15.19 | 5.59 |
| FGSM-GA | 1 | 44.98 | 21.25 | 20.75 | 20.52 | 16.46 | 18.73 |
| + QUB-decreasing | 1 | 38.39 | 21.87 | 21.57 | 21.33 | 17.21 | 19.95 |
| N-FGSM | 1 | 44.96 | 21.20 | 20.80 | 20.52 | 16.70 | 2.66 |
| + QUB-decreasing | 1 | 41.97 | 21.67 | 21.31 | 21.10 | 17.18 | 3.79 |
| FGSM-PGI(MEP) | 1 | 42.72 | 23.63 | 23.35 | 23.07 | 17.58 | 5.51 |
| + QUB-decreasing | 1 | 43.32 | **23.79** | **23.40** | **23.38** | 18.03 | 7.02 |
| ELLE-A | 1 | 43.95 | 17.62 | 16.91 | 16.34 | 13.42 | 5.98 |
| + QUB-decreasing | 1 | 41.97 | 21.67 | 21.31 | 21.10 | 17.18 | 6.96 |
| FGSM-UAP | 1 | 46.25 | 21.86 | 21.53 | 21.36 | 17.26 | 4.88 |
| + QUB-decreasing | 1 | 42.45 | 22.49 | 22.20 | 21.94 | 17.71 | 6.28 |
| PGD AT | 10 | 43.15 | 21.93 | 21.67 | 21.44 | 17.44 | 11.29 |
| + QUB-decreasing | 10 | 41.00 | 22.94 | 22.75 | 22.61 | **18.38** | 13.16 |

