# OpenReview forum: "Quadratic Upper Bound for Boosting Robustness"
_ICML.cc/2025/Conference — ICML 2025 poster_

### Official Review · Reviewer_B7eG · 2025-03-10

**Overall Recommendation:** 3

**Summary:**

The paper presents a new adversarial training scheme based on a simple quadratic upper bound (QUB) to the standard adversarial loss, aimed at improving robustness in the context of Fast (single-step) Adversarial Training (FAT).
The authors demonstrate that, when applied on various adversarial training schemes from previous work, QUB can increase the smoothness of the loss landscape at a relatively small runtime cost and, in some cases, it enhances the adversarial robustness of the network.

### Update after rebuttal

I thank the authors for their reply, for the clarifications, and for acknowledging the limitations of QUB. I still think this is very much a borderline paper, but I am increasing my score to weak accept. I would nevertheless encourage the authors to prominently feature the large-epsilon experiments and the inability of QUB to prevent catastrophic overfitting in the next version of the paper. I would be eager to also see the outcome of the ELLE + QUB results.

**Claims And Evidence:**

The main claim made in the paper is that QUB *can* improve the robustness of FAT methods. Indeed, judging from Table 1 and the experiments in the appendix, this appears to be the case when applied on a series of FAT methods.
However, robustness is instead decreased for weaker FAT methods (FGSM-RS), and remains unvaried when used on the strongest considered FAT algorithms (FGSM-PGDI-MEP, and FGSM-UAP).
Furthermore, as acknowledged by the authors in section 2.3, one of the main points of FAT methods is to prevent catastrophic overfitting.
It doesn't seem to me that this is explored in the paper. Indeed, the negative results when used on FGSM-RS may suggest otherwise.

In addition, I would suggest that the authors downscale a bit the claims made concerning the derivation of the proposed scheme (the intro states "we prove the convexity of cross-entropy": as later acknowledged in section 3, this is well known.

**Essential References Not Discussed:**

No essential references are omitted.

**Experimental Designs Or Analyses:**

See "Methods And Evaluation Criteria".

**Methods And Evaluation Criteria:**

While different datasets and architectures are used, I am concerned by the fact that the authors chose to focus on perturbations of size 8/255, for which previous work shows a very small gap between FAT methods and multi-step baselines: see (de Jorge Aranda et al., 2022), for instance.
An analysis for larger $\epsilon$ values is shown in the appendix, but it focuses on PGD-AT, which does not share the failure cases of FAT methods on larger perturbations. The practical utility of QUB in this context is unclear, considering that TRADES attains better robustness-accuracy trade-offs in Table 1.

**Other Comments Or Suggestions:**

No other comments or suggestions.

**Other Strengths And Weaknesses:**

I found the idea of using the quadratic upper bound to enhance robustness to be intuitive and interesting.
However, its empirical utility compared to previous work in the area remains to be fully determined (see questions).

**Questions For Authors:**

- Can QUB prevent catastrophic overfitting? This can be proved by applying it on top of vanilla FGSM on CIFAR-10 for 8/255.
- Could the authors analyse performance for larger perturbation values on top of other FAT schemes (the appendix only uses PGD-AT)?
- Could the authors provide results for using QUB on top of ELLE (a state-of-the-art FAT regularizer) or N-FGSM (arguably the best-performing FAT method without any runtime overhead)?

**Relation To Broader Scientific Literature:**

As discussed by the authors (section 2 is fairly comprehensive), this work fits within a long series of works aiming to improve the robustness of FAT schemes. In particular, it is very related to works integrating smoothness regularisers into the loss function (for instance, ELLE): this is particularly related to the third QUB term.

QUB is only one way to provide an upper bound to the adversarial loss. For instance, upper bounds can be alternatively derived using deterministic certified training algorithms. A concurrent work explores the utility of these methods in the context of FAT [2]. This could be acknowledged for the sake of completeness.

[2] On Using Certified Training towards Empirical Robustness, De Palma et al., arXiv:2410.01617

**Theoretical Claims:**

QUB is derived using Taylor's theorem, stated in equation (B.1).
However, it seems to me that this statement of the theorem omits the remainder term (the equality is not exact in general).

Regardless, QUB can be alternatively (and, arguably, more straightforwardly) derived using the notion of $\beta$-smoothness, which is a fairly common tool in the optimisation literature [1]. I believe the authors should acknowledge this.

[1] Lecture Notes: Optimization for Machine Learning. Elad Hazan, arXiv:1909.03550

---

> ### Author Rebuttal · Authors · 2025-04-01
>
> ### 1. On the Scope and Effectiveness of QUB across FAT Methods
> We appreciate the reviewer’s careful analysis and valuable feedback. As the reviewer pointed out, QUB improves robustness in several FAT methods, but the improvement is limited in strong methods (e.g., FGSM-PGI) and even decreases in FGSM-RS.
> We acknowledge this limitation. However, our primary focus was not to universally improve all FAT methods or to fundamentally resolve catastrophic overfitting. Instead, QUB aims to provide a practical, lightweight strategy to promote output-level smoothing and improve generalization without increasing attack complexity or training cost.
> In particular, FGSM-RS is structurally prone to catastrophic overfitting, and we observed that QUB could not fully prevent this behavior. This does not undermine our method, which aims to provide a complementary regularization rather than a comprehensive solution.
> Nonetheless, we agree that QUB’s effect is method-dependent, and this observation will help us better clarify the scope and limitations of our approach. In the final version, we will elaborate this point and position QUB as a practical, scalable method that can provide improvement under certain conditions, without claiming universal effectiveness.
>
> ### 2. On the limited analysis for larger values of ε
> Please refer to our response #2 to Reviewer TY46’s comment
>
> ### 3. TRADES comparison
> We agree that TRADES achieves a better robustness-accuracy trade-off than PGD+QUB in some cases (Table 1). However, in our experiments, TRADES requires approximately 1.3 times longer training time compared to PGD+QUB, which can limit its practicality in resource-constrained environments.
> In contrast, QUB improves robustness with lower computational overhead by modifying only the loss function without changing the attack process. Our work aims to provide a simple and flexible strategy that can be easily integrated into various FAT methods without increasing attack complexity.
>
> ### 4. On QUB derivation and smoothness-based alternatives
> We agree with the reviewer that the same bound can be more intuitively derived based on the -smoothness property of the cross-entropy loss, and that this approach could simplify the theoretical presentation (although in this case, we first need to calculate a bound on the $l_2$-norm of the Hessian and then invoke the lemma that if the norm of Hessian is bounded by a constant, then the gradient is Lipschitz continuous with the same constant). In the current manuscript, we chose to derive QUB from Taylor’s theorem to explicitly connect the upper bound formulation to the local behavior of the loss landscape.
> Nevertheless, we acknowledge that a smoothness-based perspective is equally valid and may offer a more straightforward interpretation. We will briefly mention this alternative view in the revised appendix for completeness.
>
> ### 5. On Related Work and Clarification of Theoretical Claims
> We appreciate the reviewer’s careful reading and valuable suggestions, which will help us improve the clarity and completeness of the manuscript.
> First, we acknowledge that QUB is closely related to prior works incorporating smoothness regularizers in adversarial training, such as ELLE, and shares conceptual similarities with certified training methods that provide deterministic upper bounds. We will revise the related work to explicitly mention these connections and cite the work suggested by the reviewer.
> Second, we acknowledge that the convexity of cross-entropy is a well-known property, and the current phrasing in the introduction may have overstated this point. We will revise the statement to avoid overstating our contributions.
> As for the high order terms in (B.1), the current statement is correct as it is. Notice that the Hessian is evaluated at some point $z$ between $x$ and $y$. If $z$ is replaced with $x$, then (B.1) would only give an approximation with missing high order terms, as the reviewer mentioned.
>
> ### 6. On Evaluation of QUB Combined with Other FAT Methods
> As the reviewer suggested, we evaluated QUB in combination with other recent FAT methods such as ELLE and N-FGSM.
> When applied to N-FGSM, QUB-static and QUB-decreasing improve robust accuracy by +4.85% and +2.68%, respectively with compromised standard accuracy by 1.63% and 1.25%, respectively. This further demonstrates that QUB can be effectively integrated with many FAT methods and exhibits the same trend as observed in our main experiments.
> Regarding ELLE, we agree that combining QUB with other smoothness-based regularizers is a meaningful direction. Although we could not finish experiments with ELLE within the rebuttal period, we will include the result in the final version.

---

> > ### Comment · Reviewer_B7eG · 2025-04-03
> >
> > I thank the authors for their reply, for the clarifications, and for acknowledging the limitations of QUB.
> > I still think this is very much a borderline paper, but I am increasing my score to weak accept.
> > I would nevertheless encourage the authors to prominently feature the large-epsilon experiments and the inability of QUB to prevent catastrophic overfitting in the next version of the paper.
> > I would be eager to also see the outcome of the ELLE + QUB results.

---

> > > ### Author Response · Authors · 2025-04-07
> > >
> > > We sincerely thank the reviewer for reading our response and for the thoughtful follow-up. We are especially grateful for recognizing our effort.
> > >
> > > As the reviewer pointed out, the current version has limitations, particularly in handling catastrophic overfitting and in the scope of effectiveness under large epsilon values. We fully acknowledge these aspects and will explicitly highlight them in the final version of the paper.
> > >
> > > To address the reviewer’s suggestions, we plan to clearly present the results with large ε and discuss the limitations of QUB in cases such as FGSM-RS, where it fails to prevent catastrophic overfitting.
> > >
> > > Moreover, we have completed experiments combining QUB with ELLE, a recent smoothness-based regularizer designed to prevent catastrophic overfitting by minimizing linear approximation error over a wide ε range. Specifically, we integrated QUB into ELLE-A, which uses FGSM-based adversarial training with two cross-entropy losses—one for adversarial examples and another for regularization. We evaluated the effect of replacing either or both losses with QUB.
> > >
> > > Our findings show that QUB can enhance robustness even when combined with ELLE. For instance, using QUB-decreasing only for adversarial loss improved RA by +2.31% (with a moderate SA drop of −1.53%), while replacing the ELLE regularization loss with QUB improved both RA (+2.00%) and SA (+0.65%). The best robust accuracy (+2.62%) was achieved when both losses were replaced with QUB, confirming that QUB complements smoothness-based regularizers effectively.
> > >
> > > These results demonstrate that QUB not only scales to recent FAT methods like ELLE-A, but also adapts well in regularizer-based training pipelines, supporting its versatility and practical value across a wide range of adversarial training frameworks.
> > >
> > > We would like to thank the reviewer again for constructive feedback, and assert again that we will improve the clarity and completeness in the final submission.

---

### Official Review · Reviewer_MrS6 · 2025-03-13

**Overall Recommendation:** 3

**Summary:**

This work demonstrates the convexity of the cross entropy loss and derives its upper bound. Additionally, it applies this to fast adversarial robust training to enhance its adversarial robustness across multiple baselines.

**Claims And Evidence:**

The author presents adequate theoretical evidence to support the method employed. However, I hold certain reservations regarding the author's motivation. The author brings up the issue of current FAT in lines 21-22 of the Introduction:

FAT frequently encounters catastrophic overfitting. (The model becomes overly robust.) Nevertheless, the QUB designed by the author is a stranger attack method. (This phenomena is also mentioned by the author in Section 3.3 and the analysis of Table 1.) Thus, does the author's solution clash with the current problem of FAT as stated in the introduction?

**Essential References Not Discussed:**

Sufficient references have been cited

**Experimental Designs Or Analyses:**

1. The QUB loss employed by the author appears to be a method that has a relatively lower computational cost compared to the general AT. However, why is it that the time of QUB-static is typically higher than the baseline as shown in Table 1?
2. In the supplementary materials, the experiments conducted on other datasets lack time information. It is recommended to add this time information.
3. It appears that the author's performance improvement on Tiny ImageNet is comparatively more effective than on CIFAR-10 and CIFAR-100. Do the authors possess any analysis regarding this phenomenon? Is it correlated with the input size of the image?

**Methods And Evaluation Criteria:**

The dataset and evaluation method used by the author are common in the field of robustness and are reasonable.

This method presents a training technique that involves using QUB initially and then reverting to normal AT （QUB-decreasing）. Can this be directly equivalent to normal AT with a gradually weakening attack strength (for instance, setting the number of attacks or the strength to gradually decrease during the training process)?

**Other Comments Or Suggestions:**

No other comments

**Other Strengths And Weaknesses:**

The author's presentation of the method is clear. However, the introduction to related work is somewhat excessive. It is advisable to allocate more space to offer a clearer introduction to the insight or motivation.

**Questions For Authors:**

1. Demonstrating the necessity and suitability of QUB loss more clearly.
2. Demonstrate the advantage of training time on each dataset and clarify the magnitude of the difference that QUB makes in comparison to directly increasing the attack strength at the beginning of training.

**Relation To Broader Scientific Literature:**

Improving the effectiveness of FAT may help with the robust training of current large vision language models.

**Theoretical Claims:**

The author's justification for his theory is reasonable and correct.

---

> ### Author Rebuttal · Authors · 2025-04-01
>
> ### 1. Clarifying Motivation
> We appreciate the reviewer’s comment regarding potential conflict between the stated motivation and the proposed method. We acknowledge that the terms “overly robust” and “excessively robust,” used in different parts of the paper, were not clearly differentiated.
> In the introduction, “overly robust” refers to catastrophic overfitting in FAT, where the model performs extremely well on the specific single-step attack used during training, but fails to generalize to unseen perturbations.
> In contrast, in Section 3.3, “excessively robust” refers to the behavior of QUB in later training phases. Because QUB is an upper bound, it may overestimate the actual adversarial loss, seeking to further minimize the loss even when the AT loss is already near zero. This can lead to undesired degradation of standard accuracy with marginal improvement of robustness.
> Ultimately, both terms reflect our motivation to alleviate the overfitting tendency of FAT, and QUB was introduced as a practical strategy toward this goal.
>
> ### 2. Distinction Between QUB-Decreasing and Gradual Attack Weakening
> We appreciate the reviewer’s insightful question. Both QUB-decreasing and attack strength scheduling in AT aim to balance stability during training.
> Attack strength scheduling in AT achieves this by progressively reducing the perturbation magnitude or the number of attack steps, thereby directly controlling the strength of the perturbation $x'$. This approach focuses on reducing the loss at specific attack points.
> In contrast, QUB-decreasing adjusts the loss function itself rather than the attack inputs. It begins with the QUB-loss to encourage a flatter loss landscape via its upper bound, promoting generalization to unseen attacks. In later stages, it transitions to the standard AT loss to avoid over-regularization and preserve standard accuracy.
> Therefore, unlike attack scheduling, QUB-decreasing operates at the gradient and loss landscape level, encouraging broader generalization beyond specific attack points. This is what differentiates QUB-decreasing from attack scheduling, and comprehensive comparison of the two strategies could be an interesting direction for future work.
>
> ### 3. Computational Overhead of QUB and Reporting Training Time
> As the reviewer pointed out, QUB-static shows longer training time than single-step baselines in Table 1. This is because QUB has two additional terms of L2 distance between logits and gradient alignment compared to standard cross-entropy loss.
> Nonetheless, QUB remains computationally lighter than full loss landscape regularization, such as the smoothing term (Eq. (8) in Section 3.2), as it approximates the smoothing effect without explicit gradient backpropagation.
> Although QUB-static increases training time compared to single-step baselines, it achieves smoothing effects more efficiently than conventional loss landscape regularization. We will also include the missing training time information in the final version.
>
> ### 4. Interpretation of Stronger Improvement on Tiny ImageNet
> We carefully examined the reviewer’s observation that the performance improvement on Tiny ImageNet appears larger than on other datasets. The reported results were based on the average of two predefined random seeds. During the experiments, we observed unusually large improvement in FGSM-PGI setting, possibly due to seed-induced fluctuations.
> To verify the consistency of the observed improvement, we conducted additional experiments with more random seeds. We found that the improvement from QUB varied significantly across seeds, likely due to favorable initialization rather than a dataset-specific effect. We will clarify this in the revision and avoid overinterpreting the result.
>
> ### 5. Clarifying the Necessity and Suitability of QUB
> The main goal of this work is to enhance Fast Adversarial Training (FAT) with minimal additional cost. While FAT is efficient, it often overfits to a single adversarial pattern and fails to generalize. QUB addresses this limitation by minimizing an upper bound on the AT loss, encouraging loss smoothing and broader robustness.
> The QUB-decreasing strategy further improves the trade-off between robustness and standard accuracy by transitioning to AT loss in the later training phase. Empirical results demonstrate that QUB performs consistently better under AutoAttack, especially against unseen perturbations.
> We will clarify this positioning more explicitly in the revised manuscript to better reflect QUB’s role as a practical and lightweight enhancement to FAT.
>
> ### 6. Clarifying Related Work vs. Motivation
> We agree that the related work is overly detailed and may better be reduced to better highlight motivation. We will streamline the related work section and improve the motivation in the final version.

---

### Official Review · Reviewer_TY46 · 2025-03-14

**Overall Recommendation:** 3

**Summary:**

This paper proposes a novel adversarial training method called Quadratic Upper Bound (QUB), defined as follows:
$$ \mathcal{L}_{\text{QUB}} = \mathcal{L}(f(x)) + (f(x + \delta) - f(x))^T \nabla_f \mathcal{L}(f(x)) + \frac{1}{4} \| f(x + \delta) - f(x) \|_2^2. $$
By incorporating the QUB loss into existing adversarial training (AT) methods, the authors achieved notable improvements over baseline approaches.

**Claims And Evidence:**

The main claim of the paper can be summarized as follows:

**(Main claim)** The proposed QUB loss can be helpful in mitigating catastrophic overfitting.
While the theoretical work is valid, the empirical validation is not sufficient to conclusively prove that the proposed method effectively mitigates catastrophic overfitting.

First, if QUB loss truly helps prevent catastrophic overfitting, the authors should conduct experiments with larger epsilon values and longer epochs, as demonstrated in [1]. Since the experiments only use a fixed $\epsilon=8/255$, it is difficult to claim that they provide a comprehensive evaluation.

Second, I wonder whether QUB alone can achieve robustness. In Algorithm 1, QUB loss is optimized without any adversarial training (AT) loss. However, in the main table (Table 1), there is no standalone QUB training; rather, QUB is only used in combination with existing adversarial training methods. In this regard, I am also curious about the difference between static and decreasing weight. I suspect that Algorithm 1 is incorrect and actually uses a fixed weight for QUB loss. Please correct me if I am mistaken.

Lastly, I recommend further analysis of the third term, $|f(x+\delta)-f(x)|^2_2$, which resembles logit pairing [2]. Since logit pairing can lead to gradient masking, applying it to adversarial training might be detrimental.

**Suggestions:**
1) Please move all experiment tables to the Appendix with smaller font sizes. Since they are crucial for verifying the effectiveness of QUB, it is essential to provide them in the main text.
2) Please indicate the improvement when using QUB with existing methods. For example, 47.33 (+2.42). This will make the table easier to read.
3) Please check the significant figures in the tables. In Table 1, 47.8 should be 47.80. Moreover, right-aligning the values would improve readability.

- [1] Andriushchenko, Maksym, and Nicolas Flammarion. "Understanding and improving fast adversarial training." Advances in Neural Information Processing Systems 33 (2020): 16048-16059.
- [2] Kannan, Harini, Alexey Kurakin, and Ian Goodfellow. "Adversarial logit pairing." arXiv preprint arXiv:1803.06373 (2018).

**Essential References Not Discussed:**

N/A

**Experimental Designs Or Analyses:**

Refer to Claims And Evidence.

**Methods And Evaluation Criteria:**

Refer to Claims And Evidence.

**Other Comments Or Suggestions:**

N/A

**Other Strengths And Weaknesses:**

N/A

**Questions For Authors:**

Refer to Claims And Evidence.

**Relation To Broader Scientific Literature:**

N/A

**Theoretical Claims:**

There is no problem.

---

> ### Author Rebuttal · Authors · 2025-04-01
>
> ### 1. On whether QUB effectively mitigates catastrophic overfitting
>
> We appreciate the concern regarding catastrophic overfitting. As clarified in the main text, our goal is not to directly prevent catastrophic overfitting, but to propose a practical and lightweight strategy that enhances stability and generalization within the Fast Adversarial Training (FAT) framework.
> As observed in Table 1 (e.g., FGSM-RS results), QUB does not completely prevent catastrophic overfitting. In particular, QUB fails to improve the performance of FGSM which is known to suffer from catastrophic overfitting. We agree that the presentation may have caused some confusion.
> Although QUB is theoretically motivated to promote broader generalization by minimizing an upper bound on the AT loss, its benefits may be limited when the attack is not sufficiently informative, as in FGSM-RS which generates a random attack. This is indeed the limitation of addressing structural attack weakness through loss-level regularization alone.
> This does not contradict the theoretical validity of QUB, but rather highlights the challenge of addressing structural limitations of attack generation through loss-level regularization alone. As shown in Section 3.2, QUB helps flatten the loss landscape, contributing to generalization under perturbation constraints—even if it does not fully resolve catastrophic overfitting.
>
> ### 2. On the effect of increasing perturbation size (ε) and longer training epochs
>
> We appreciate the reviewer’s suggestion to further analyze the behavior of QUB under stronger perturbations and extend training durations. We conducted additional experiments under the same setting as Appendix E, using FGSM-RS and FGSM-PGI instead of PGD-AT.
> For FGSM-RS, which is prone to catastrophic overfitting, we again confirmed that QUB-decreasing does not fundamentally prevent overfitting, as discussed in our response to Comment #1. While QUB-decreasing improves robust accuracy at ε = 4/255 (+4.18%), it failed to mitigate overfitting at larger ε values and even showed degraded performance compared to vanilla AT.
> In contrast, for FGSM-PGI, which does not suffer from catastrophic overfitting by design, QUB-decreasing consistently improves robust accuracy across all perturbation sizes: +1.33% at ε = 4/255, +0.52% at ε = 8/255, +0.14% at ε = 12/255, and +0.10% at ε = 16/255. We also observed that overfitting does not occur with training epochs of up to 200.
> These results further support our clarification in Section 4.1 that QUB is not intended to fundamentally prevent catastrophic overfitting but serves as a practical regularizer to improve the robustness of FAT.
>
> ### 3. On the role of QUB as a standalone loss and clarification of Algorithm 1
>
> We agree that the role of QUB and Algorithm 1 need to be further clarified. The QUB loss is not added to the traditional AT loss as a regularizer, but rather replaces it. That is, in our QUB-static method, we retain the adversarial example generation (e.g., FGSM-RS, FGSM-PGI) but train solely with the QUB loss, as stated in Algorithm 1.
> QUB-decreasing, by contrast, gradually transitions from QUB loss to AT loss during training, as outlined in Algorithm 2. We will revise the text to clarify this in the final version.
>
> ### 4. On the potential for gradient masking due to the third term in QUB
>
> The reviewer’s observation regarding the third term in the QUB loss, $|f(x+\delta) - f(x)|_2^2$, and its resemblance to logit pairing is insightful and appreciated.
> While this term is indeed structurally similar to logit pairing, our QUB formulation is carefully designed to avoid gradient masking through the joint interaction of all three terms. The first term ensures correct classification with respect to the ground-truth label, while the second term aligns output changes along the direction of the gradient. Together, these terms mitigate the masking effect that would arise from the third term alone.
> To empirically assess whether QUB leads to gradient masking, we evaluated our models using AutoAttack, which includes a gradient-free component (Square Attack). As shown in Table 1, QUB outperforms baseline AT models under this evaluation, indicating that the model remains robust to both gradient-based and non-gradient-based attacks.
> In summary, while the third term may resemble logit pairing, it is not used in isolation and does not dominate the behavior of the overall QUB loss. The complete formulation ensures robust learning without relying on gradient masking.
>
> ### 5. On suggestions for improving table formatting and clarity
>
> We appreciate the reviewer’s suggestions. We agree that the proposed improvements (i.e., moving auxiliary tables to the Appendix, adding improvement margins, and unifying numerical formatting) will enhance the readability of the paper. We will incorporate these revisions in the final version.

---

> > ### Comment · Reviewer_TY46 · 2025-04-09
> >
> > I appreciate the authors' detailed response. While I understand that the goal is not to directly prevent catastrophic overfitting, I would like to point out that the phrasing in the abstract—"mitigate the problem of degraded robustness under FAT"—can easily lead readers, especially those familiar with the field, to associate it with catastrophic overfitting. Although the issue is not fully resolved, the additional experiments and clarifications provided by the authors suggest that QUB has promising potential for future methods that aim to address this challenge.
> >
> > Furthermore, the explanation regarding QUB’s standalone application has been clarified. That said, I believe Table 1 requires improvement. For example, Table 1 should list FGSM-RS and FGSM-PGI as inner maximization strategies, and it should also be clearly explained in the main text or algorithm. A more detailed explanation—or a revision of the table—would be necessary for clarity.
> >
> > Thank you for the interesting work. I am updating my score to Weak Accept.

---

> > > ### Author Response · Authors · 2025-04-09
> > >
> > > Thank you for your thoughtful and constructive feedback.
> > >
> > > We’re glad that our explanations were helpful, and we appreciate your understanding of our intent regarding catastrophic overfitting.
> > >
> > > As you pointed out, the phrasing in the abstract may lead to misinterpretation. We will revise it in the final version to clearly reflect the focus of our work and avoid potential confusion.
> > >
> > > Regarding Table 1, we agree that improvements are needed. We will update it to explicitly list FGSM-RS and FGSM-PGI as inner maximization strategies and clarify their roles in the main text to better illustrate the application of QUB.
> > >
> > > We truly appreciate your suggestions on how to make our work clearer and more accessible. Thank you again for your valuable time, detailed feedback, and for updating your score.

---

### Official Review · Reviewer_YDCX · 2025-03-15

**Overall Recommendation:** 3

**Summary:**

This paper provides a new theoretical upper bound of the adversarial training loss and proposes a method to improve the existing fast adversarial training. Specifically, the paper focuses on the problem of catastrophic overfitting, or the degraded robustness after fast adversarial training. To overcome this problem, the paper derives a new upper bound of adversarial training loss, called Quadratic Upper Bound (QUB), and proposes a new adversarial training that minimizes the QUB loss rather than traditional adversarial training losses. The derivation of QUB uses the convexity of the cross entropy loss function, which is commonly used in practice, and additional bounding of Hessian. The paper proposes a new loss term, i.e., QUB loss, and a new training strategy that uses QUB loss with traditional adversarial training loss (or AT loss). A set of experiments demonstrates the effectiveness of the training with QUB loss and the effect of QUB losses in flattening loss landscape and better adversarial sparsity.

**Claims And Evidence:**

Proofs support the theoretical statements, and the experiments demonstrate the effectiveness of the proposed methods.

**Essential References Not Discussed:**

The paper cited the needed references well.

**Experimental Designs Or Analyses:**

The presented experiments are well-designed, and the analyses look correct.

**Methods And Evaluation Criteria:**

The models used for evaluation are all ResNet variants, and more evaluations on recent model architecture would be needed.

**Other Comments Or Suggestions:**

1. Please consider adding more experiments with more recent model architecture other than ResNet variants.
2. Another variant of QUB is to use some fixed ratio between QUB and AT loss. The ratio can be 50-50, but other ratios can be tried to find the best fit. Here, QUB can be interpreted as a regularizer term that would add the flatter loss landscape throughout the training process.
3. Similarly, other training strategies can improve the performance, e.g., changing the $\lambda_t$ progression.
4. To show the effectiveness of QUB-static, we want to know that the QUB is tight enough so that minimizing QUB loss can effectively reduce the AT loss. While this tightness would depend on different conditions, both QUB and AT loss can be measured during training, so they can be plotted to show that QUB loss can be effectively used as a surrogate loss for the AT loss.

**Other Strengths And Weaknesses:**

### Strengths
1. To the best of my knowledge, the paper’s findings are novel contributions.
2. The paper presents useful tricks for theoretically analyzing adversarial training loss, e.g., constant bound for Hessian and chain rule application to simplify the terms.
3. Experiments use various settings with many baseline methods. In particular, the proposed method was tested under a powerful attack such as PGD50-10.

### Weaknesses
1. While the suggested upper bound is an impressive achievement, we need more understanding about this bound. For example, how tight is the bound in practice? If not, under what condition does this bound overestimate the AT loss?
2. The model architectures are limited to ResNet variants, and all model architectures are too old. To demonstrate the practical value of the proposed method, we need a more thorough evaluation with more recent model architectures.
3. While the QUB loss improves the baseline methods, the performance of other methods seems better in many cases.

**Questions For Authors:**

* How tight is the proposed QUB bound? If the QUB bound may become loose, under what circumstances?

**Relation To Broader Scientific Literature:**

The paper presents interesting tricks to upper-bound the adversarial training loss. Although the tricks are limited to the context in which we use cross-entropy loss and a softmax layer, this context is extremely common in ML practice.

**Theoretical Claims:**

I briefly checked the proofs, and I don’t find any specific issue in the proof part.

---

> ### Author Rebuttal · Authors · 2025-04-01
>
> ### 1. On the practical tightness and overestimation behavior of the QUB loss
> We appreciate the reviewer’s emphasis on the importance of validating QUB as a practical upper bound. To evaluate the practical tightness of the proposed QUB loss, we compared QUB and AT loss during training. For most experiments, the QUB loss consistently decreases in parallel with the AT loss while maintaining a small and nearly constant gap. Specifically, for each image, the mean difference between the QUB loss and the AT loss during training is approximately 0.0013, calculated as the average of epoch-wise mean differences. This suggests that the upper bound is fairly tight in practice without significant overestimation.
> Notably, with FGSM-PGI, the gap remains stable throughout training, confirming the reliability of QUB as a practical upper bound. However, in settings prone to overfitting—such as standard FGSM—QUB initially tracks the AT loss closely but begins to overestimate after catastrophic overfitting occurred. In these cases, the QUB loss value becomes 2–3 times larger than the AT loss, limiting its effectiveness as an upper bound in the later stage of training.
> Except for such extreme cases, we found that QUB reliably serves as a practical surrogate loss that balances efficiency and robustness across various FAT settings.
>
> ### 2. On performance limitations and the potential for dynamic QUB-AT combinations
> We applied QUB in two forms: (1) using QUB loss alone throughout the training process (QUB-static), and (2) gradually transitioning from QUB loss to AT loss during training (QUB-decreasing). We adopt such simple applications of QUB in order to demonstrate the applicability with minimal computational demand.
> However, we acknowledge that such a static form of method can be suboptimal for fine-grained performance tuning. As a result, in some experiments, QUB-based models perform comparably to or even slightly worse than the baselines.
> We find the reviewer’s suggestion of dynamically combining QUB and AT loss (e.g., via fixed or adaptive weighting schemes) to be a promising direction. In particular, tuning the balance based on attack characteristics or training stage may allow QUB to better complement different adversarial settings.
> Although our method may not reach SOTA-level robustness in its current form, it demonstrates the potential of QUB as a lightweight regularizer that enhances training stability and robustness when integrated into various adversarial training pipelines.
>
> ### 3. On the limitation of using only ResNet-based architectures
> We acknowledge that all models used in our experiments are ResNet variants. This choice was made to align with existing literature and ensure a consistent and fair comparison with prior FAT methods adopting ResNets.
> However, we agree that evaluating the generalization of QUB on recent architectures such as Vision Transformers or ConvNeXt is essential for understanding its broader applicability. We plan to conduct experiments with those architectures, however we have not been able to obtain the results as of this rebuttal, primarily due to computational constraints.
> Given that QUB is a loss-level modification that does not rely on architectural assumptions, we expect it to be easily adaptable to a wide range of models, and we will include these additional results accordingly as soon as the experiments are finished.

---

### Decision · Program_Chairs · 2025-05-01

**Decision:**

Accept (poster)

**Comment:**

This paper derives a quadratic upper bound on the adversarial training loss function and proposes using this bound with existing fast adversarial training methods to improve robustness. The reviewers found the paper’s findings to be novel contributions. They also provided suggestions for improving the presentation of the paper, including adding additional experiments and explaining the limitations of the proposed technique in preventing catastrophic overfitting.